# Decomposing Complex Visual Comprehension into Atomic Visual Skills for Vision Language Models

**Hyunsik Chae**[†], **Seungwoo Yoon**[†], **Chloe Yewon Chun**[†],
**Gyehun Go**[†], **Yongin Cho**[†], **Gyeongmin Lee**[†], **Ernest K. Ryu**[⋆]

[†]Seoul National University, [⋆]UCLA, Department of Mathematics
https://github.com/Atomic-Visual-Skills/AVS

## Abstract

Recent Vision Language Models (VLMs) have demonstrated impressive multi-modal comprehension and reasoning capabilities, but they often struggle with trivially simple visual tasks. In this work, we introduce the Atomic Visual Skills Benchmark (AVSBench) to evaluate whether VLMs possess capabilities to understand basic geometric features, which we refer to as atomic visual skills. Specifically, we systematically categorize the atomic visual skills and handcraft a set of 5,073 diverse questions designed to assess each individual atomic visual skill. Using AVSBench, we evaluate the current leading VLMs and find that they struggle with most of these atomic visual skills that are obvious to humans.

## 1   Introduction

Recent Vision Language Models (VLMs), also referred to more generally as Multimodal Large Language Models (MLLM), integrate vision components into language models and demonstrate an impressive breadth of multimodal comprehension and reasoning capabilities [7]. At the same time, however, VLMs often struggle with trivially easy visual tasks as shown in Figure 1, a puzzling phenomenon that seems almost contradictory to their remarkable performance [11, 43]. We propose two hypotheses to explain the observed shortcomings of current vision-language models:

*Hypothesis 1: The comprehension of complex visual diagrams requires the composition of smaller atomic visual skills.*

*Hypothesis 2: Current vision language models are incapable of such atomic visual skills.*

In this work, we introduce the Atomic Visual Skills Benchmark (AVSBench) to test *Hypothesis 2*. AVSBench is designed to rigorously evaluate VLMs' ability to comprehend fundamental geometric features, which we refer to as *atomic visual skills*. We systematically categorize 36 atomic visual skills that encompass diagrams arising in high school-level geometry and *handcraft* a set of 5,073 diverse questions designed to assess the understanding of the individual atomic visual skills.

We then evaluate the state-of-the-art VLMs on AVSBench, and the results clearly support *Hypothesis 2*. While our problems are designed to be trivial to humans, VLMs struggle; state-of-the-art models like Gemini-1.5-pro and GPT-4o score around 70%-75% on problems with the "easy" categorization, score around 60% on the "medium" problems, and 30% on "hard" problems. The confirmation of *Hypothesis 2* also lends support to *Hypothesis 1*, and suggests a promising direction of future work of training vision language models specifically on atomic visual skills to improve their performance in comprehending complex visual diagrams.

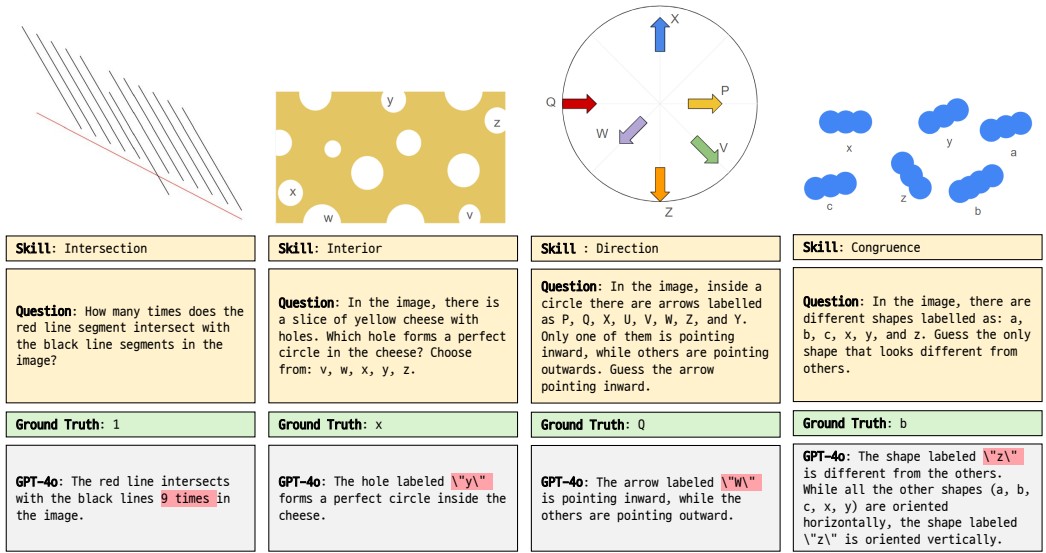

Figure 1: Examples of AVSBench problems and responses by GPT-4o. Other state-of-the-art models exhibit similar failures. These examples demonstrate a deficiency in the VLMs' understanding of basic geometric concepts.

## 2 Atomic Visual Skills Benchmark (AVSBench)

Many visual reasoning tasks in existing benchmarks, such as the ones listed in Section A, are composite tasks that can be broken down into more elementary components. This observation leads us to define a set of *atomic visual skills* based on the following criteria: (i) each skill is intuitive and trivial for adult humans, (ii) each skill cannot be decomposed further, or doing so would be unnatural, and (iii) the list of atomic visual skills should comprehensively cover the abilities required for comprehending geometric diagrams arising in high-school level mathematics. While this definition is not a fully rigorous one, we found it to be sufficiently clear and substantive for our work.

Using these criteria, we identified 36 atomic visual skills, including the ability to understand concepts such as `angle`, `boundary`, `orthogonality`, `curvature`, and `direction`. The complete list and further illustrations are provided in D.

For adult humans, these skills are trivially simple and require little to no reasoning to perform. Therefore, we use the term ***comprehension*** instead of ***reasoning*** to emphasize our belief that these skills do not require much reasoning or thinking to perform, for both humans and VLMs. This belief is partially supported by Findings 3 of Section 3.1.

We then constructed the Atomic Visual Skills Benchmark (AVSBench) to evaluate VLMs' ability to perform the 36 atomic visual skills. AVSBench, as summarized in Figure 2, comprises 5,073 new handcrafted image-question-answer triplets with the following characteristics:

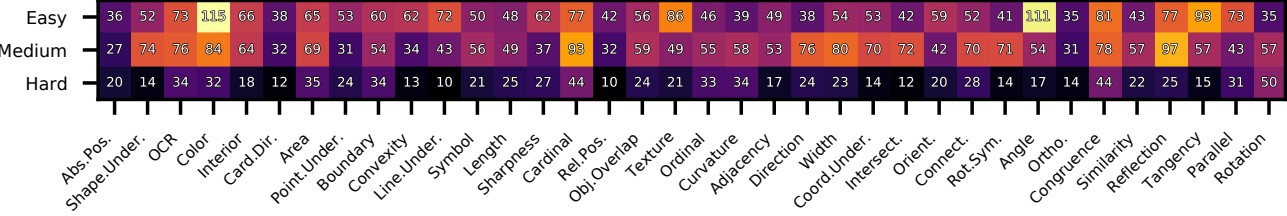

Figure 2: List of 36 atomic visual skills and the number of easy, medium, and hard problems for each skill. The difficulty is judged by the authors. We provide a total of 5,073 new handcrafted problems.

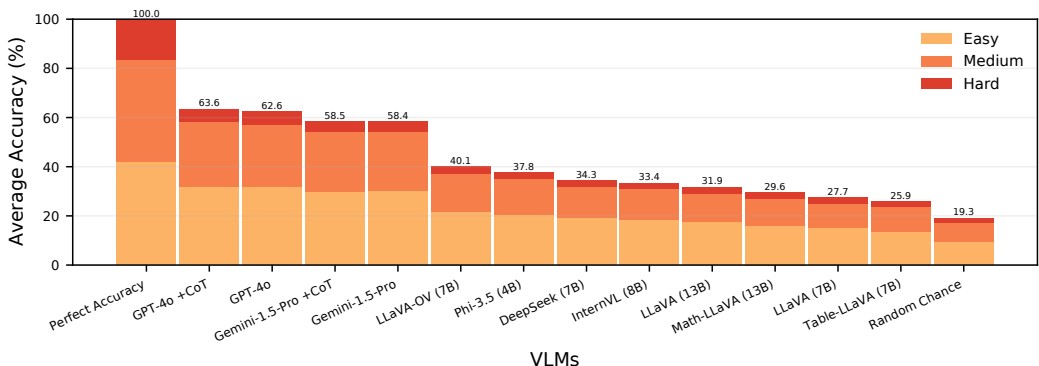

Figure 3: Evaluation results on AVSBench. *+CoT* implies the performance of the model on the right with chain-of-thought (CoT) prompting [22]. The area ratios of each colored section are aligned with the actual ratio of problem counts. Details about the models including their full name are in E. Full quantitative results are illustrated in Table 5.

- **Originality.** All images and questions are newly generated, ensuring that they are free from data contamination concerns.

- **Diversity.** Although we focus on the set of only 36 skills, the problems feature diverse expressions and formats, as illustrated by the sample problems in C.

- **Skill isolation.** Each question targets a specific atomic skill, minimizing the overlap with other skills. Recognizing the impossibility of achieving complete isolation, our method incorporates diverse tasks to mitigate the influence of any task or their relevant overlapping skills. For instance, to minimize the influence of other skills while evaluating the `cardinal` skill, we asked about cardinals of various concepts and objects, from colors to points, lines, and other figures.

- **Focus on high school geometry.** We focus on the visual skills required to solve high school-level geometry problems for the following reasons: (i) the scope of the high school mathematics curriculum is more or less clearly defined, (ii) (as our results of Section 3 show) these atomic visual skills are sufficiently challenging for current VLMs, (iii) the range of skills is broad enough to be applicable to other visual comprehension tasks, such as interpreting charts, tables, and scientific or mathematical figures.

## 3 Current vision language models struggle with atomic visual skills

We evaluate three types of VLMs on AVSBench: (i) state-of-the-art proprietary models: GPT-4o [40, 39] and Gemini-1.5-pro [49], (ii) popular mid-sized open-weight models: LLaVA-Next (7B, 13B) [30], LLaVA-OneVision (7B) [26], Phi-3.5-Vision (4B) [1], InternVL2 (8B) [10], Deepseek-VL (7B) [31], and (iii) VLMs specifically trained for geometry or other visual data: Math-LLaVA (13B) [47], Table-LLaVA (7B) [60]. Further details of model versions are provided in Section E.

The evaluation protocol consists of three steps. First, we provide the VLM with the image-question pair and solicit a response. As we further discuss later, we also explore the effect of the chain-of-thought (CoT) prompting [54, 22]. Second, we extract the answer from the VLM's response using GPT-4o mini [40]. Third, we ask GPT-4o mini to score the answer by comparing the extracted answer with the answer key. We award 1 point for a correct answer and 0 points otherwise without any partial credit. More details on our evaluation protocol are provided in F.

### 3.1 Experimental results and findings

Figure 3 presents the results comparing the selected VLMs and the baseline corresponding to random guessing. Details including exact values are provided in G. On "easy" problems, models with about 10B parameters score between 32.5% and 51.0% while closed-source models, including GPT-4o and Gemini-1.5-pro, achieve over 70%, far above random chance (22.4%). On "medium" problems, models with about 10B parameters score between 23.8% and 37.8%, slightly above random chance

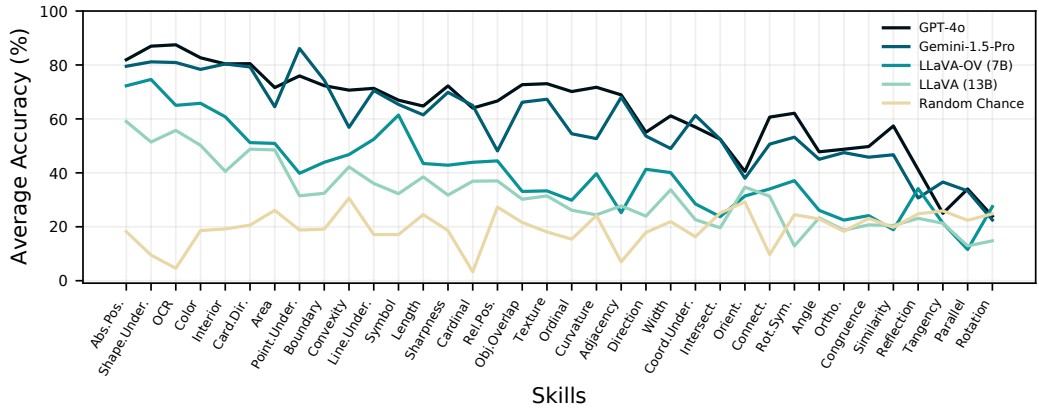

Figure 4: Accuracies of a leading model, 3 outstanding models, and random chance on each skill. The skills are ordered in descending order of accuracy, averaged over all models.

(19.1%). Closed-source models achieve between 58.6% and 64.6%. For "hard" problems, most open models score close to random chance (11.7%). The closed-source models GPT-4o (32.3%) and Gemini-1.5-pro (26.9%) scored significantly better than random chance but clearly struggled.

**Findings 1: Models share strengths and weaknesses.** Figure 4 presents the accuracies of selected models on each skill. The performances across skills varied significantly. For example, most VLMs performed well on OCR, Absolute Position, and Shapes, but performed poorly on tangency, parallel, and angle. Interestingly, the different models largely shared the same set of skills they did well on and the same set they found challenging.

**Findings 2: Domain-specific models are not better.** Surprisingly, Math-LLaVA [47] and Table-LLaVA [60], which are VLMs specifically trained for geometry or visual data, did not perform better than general VLMs of similar size, on almost any skills within AVSBench as the results of Table 5 show.

**Findings 3: Chain-of-thought is not helpful in enhancing atomic visual skills.** We also evaluated the best-performing models—GPT-4o and Gemini-1.5-pro—with chain-of-thought (CoT) prompting [22] on AVSBench. Surprisingly, CoT did not help for most skills, and for some skills, it even worsened the performance, as shown in Table 5. This contrasts with prior work, which found CoT to be beneficial for certain visual reasoning tasks [32, 53]. We attribute this difference to our hypothesis that the atomic visual skills of AVSBench require simple "comprehension" and, therefore, do not benefit from the additional "reasoning" steps afforded by CoT prompting. More concrete comparison, see G.

## 4 Conclusion

We present the Atomic Visual Skills Benchmark (AVSBench), a benchmark designed to rigorously evaluate VLMs' ability to perform atomic visual skills in a decomposed manner. We then show that current state-of-the-art VLMs struggle with such atomic visual skills.

The failure of VLMs to carry out such simple atomic visual tasks raises the question: How is it that VLMs are sometimes successful at performing complex visual tasks? For this, we hypothesize that the existing impressive performance on complex tasks is due to overfitting or unimodal shortcuts. Indeed, recent studies such as [11, 18, 57, 50] report that VLMs tend to depend on language shortcuts, as we further reference and discuss in Section A of the appendix.

Recall that our *Hypothesis 1* posits that the atomic visual skills are necessary subcomponents for comprehending complex visual diagrams. In future work, we plan to train and fine-tune VLMs directly on the atomic visual skills and ascertain *Hypothesis 1*.

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

# A    Prior works

**VLM benchmarks and language shortcuts.**    Existing VLM benchmarks evaluate models on their ability to solve diverse vision-language problems from general real-world tasks [3, 13, 15], tasks that require specific skills such as high-school geometry [32, 20, 9, 8], analyzing charts and tables [34, 60, 35], and other scientific visual data [19, 21]. However, most VLM benchmarks do not contain a mechanism for verifying whether a correct solution is based on correctly comprehending the visual information, allowing the models to sometimes rely on linguistic biases to find a solution [7]. Lin et al. [27] revealed that by simply avoiding implausible or less fluent sentences, blind language models can distinguish the correct description of an image from wrong ones on CREPE [33], VL-Checklist [59], and ARO[56]. Mathverse [57] observed that, when solving geometry problems, VLMs rely mostly on textual inputs without correctly interpreting diagrams.

Some recent work has started to seek unbiased ways to measure visual capabilities. Winoground [50] prevents choosing image captions based on the plausibility of the sentence structure, by providing two images with same objects or concepts but with different relationships. Blink [11] and CV-Bench [51] present novel vision-oriented tasks with minimized effects of linguistic biases.

**Compositional reasoning.**    There has been intensive recent research on the compositional capabilities of Language Models [4, 55, 17, 48, 44, 58, 23, 38, 42]. VLMs have additionally shown compositional capabilities in visual tasks [25, 37, 33, 59, 56]. However, such studies left the visual portion with less attention, thus vulnerable to linguistic shortcuts such as removing grammatically wrong setnences or choosing more realistic sentences as answers. To mitigate this issue, Sugar-Crepe [18] generated sentences with ChatGPT to provide incorrect captions of given images, with different compositional structures while as realistic as the ground truths.

**Research on atomic skills of LLMs.**    To understand the capabilities of LLMs, there has been prior work on studying LLMs in simple idealized experiments. This includes research on in-context learning [28, 36], arithmetic (addition and multiplication) [38, 16, 24], fact search and reverse fact search [2, 6, 12], and programming [5, 45, 14].

However, there have been far fewer studies of this kind for vision language models. Paiss et al. [41] focused on counting objects in image and suggested CountBench. Shen et al. [46] suggest a skill-based approach to evaluating VLMs, but their list of skills is not atomic. CV-Bench [51] evaluates 4 vision-centric skills: spatial relationship, object count, depth order, and relative distance. MMVP [52] challenges VLMs to understand 9 visual patterns.

BlindTest [43] observed failures of VLMs with 7 simple tasks focusing on fundamental geometric features, some of which share similar approaches with AVSBench. While these tasks are novel and effective, they lack diversity in color, shape, or word choice. Their Task 1 for instance, uses only one red and one blue line, each with exactly one sharp turn. For Task 3, they adopted only three types of strings and a red circle to generate visual context. Moreover, these 7 tasks are insufficient to collectively evaluate the full spectrum of visual capabilities, leading to limited scope in evaluation objectives. In contrast, our AVSBench offers a systematic, comprehensive framework for evaluating a comprehensive set of visual skills of VLMs with an extensive dataset that is rich in color, shape, and other variations.

# B    Problem difficulty categorization

The problems are categorized into three difficulty levels: easy, medium, and hard. Problems categorized as easy or medium should be quickly solvable by humans, whereas hard questions, although more time-consuming, are designed to be clear and easily verifiable. We clarify that the difficulty levels were determined by the authors, so there is a degree of subjectivity to the categorization.

# C   Sample problems

We present 99 sample problems to provide the readers with the general characteristics of our AVS-Bench dataset, along with the responses of Gemini-1.5-pro[49], GPT-4o[40], and LLaVA-Next-13B[29]. For readability, we report a summary rather than the full text of the model response. See Figure 5 for examples of full model responses.

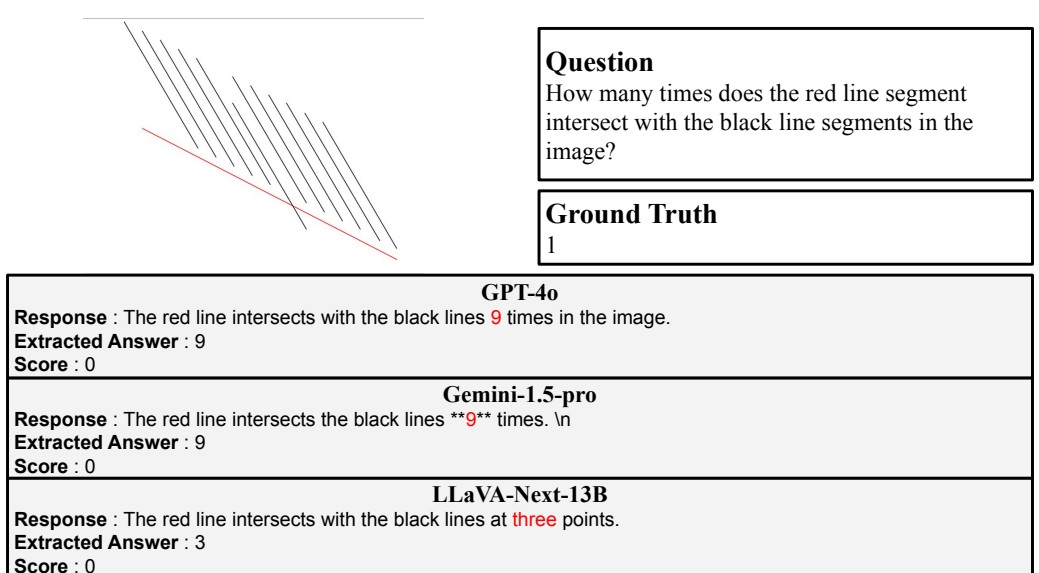

Figure 5: Full responses of VLMs and scoring by GPT-4o mini.

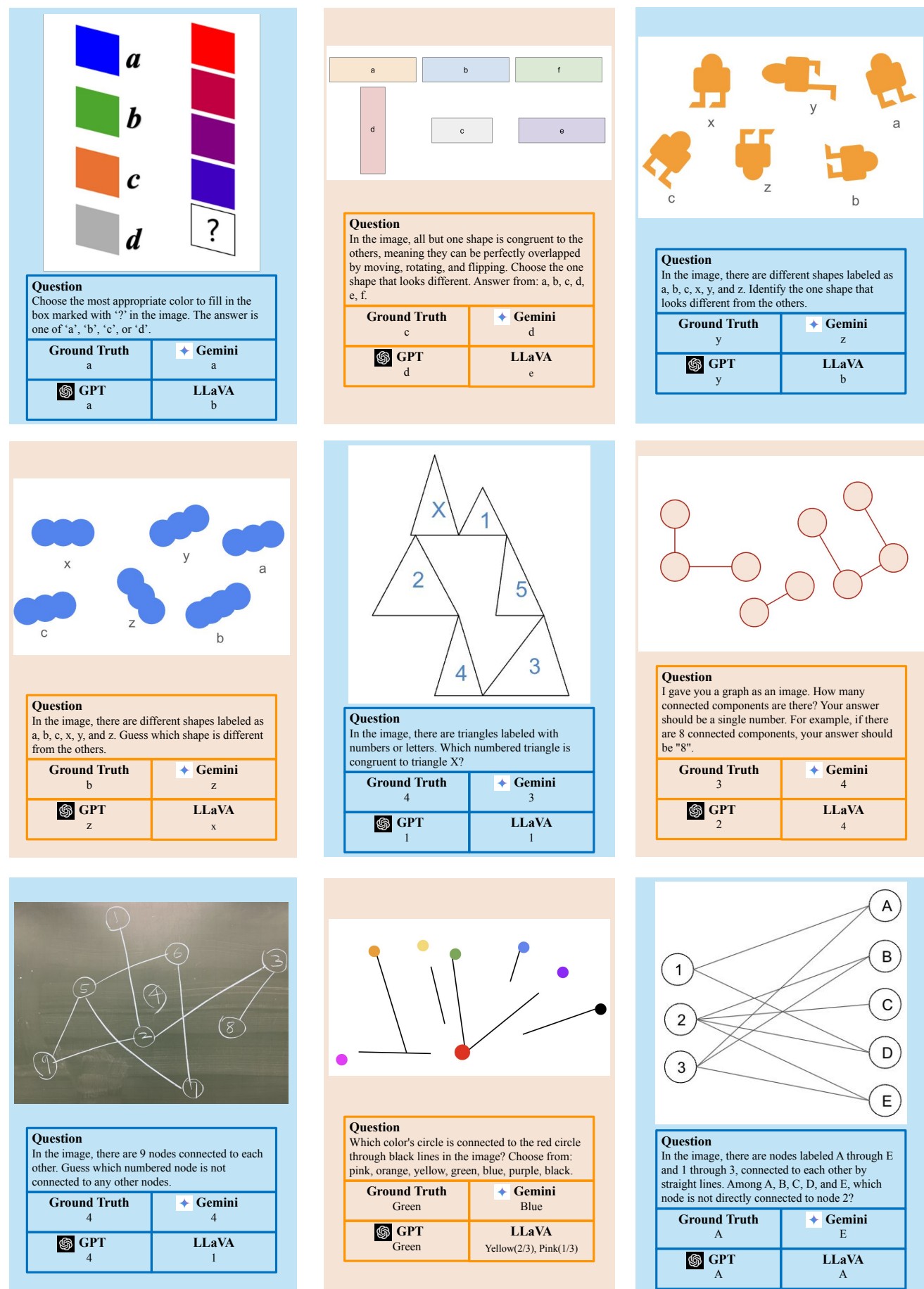

Figure 6: Sample problems from AVSBench and responses from Gemini-1.5-pro, GPT-4o, and LLaVA-Next-13B, 1/11.

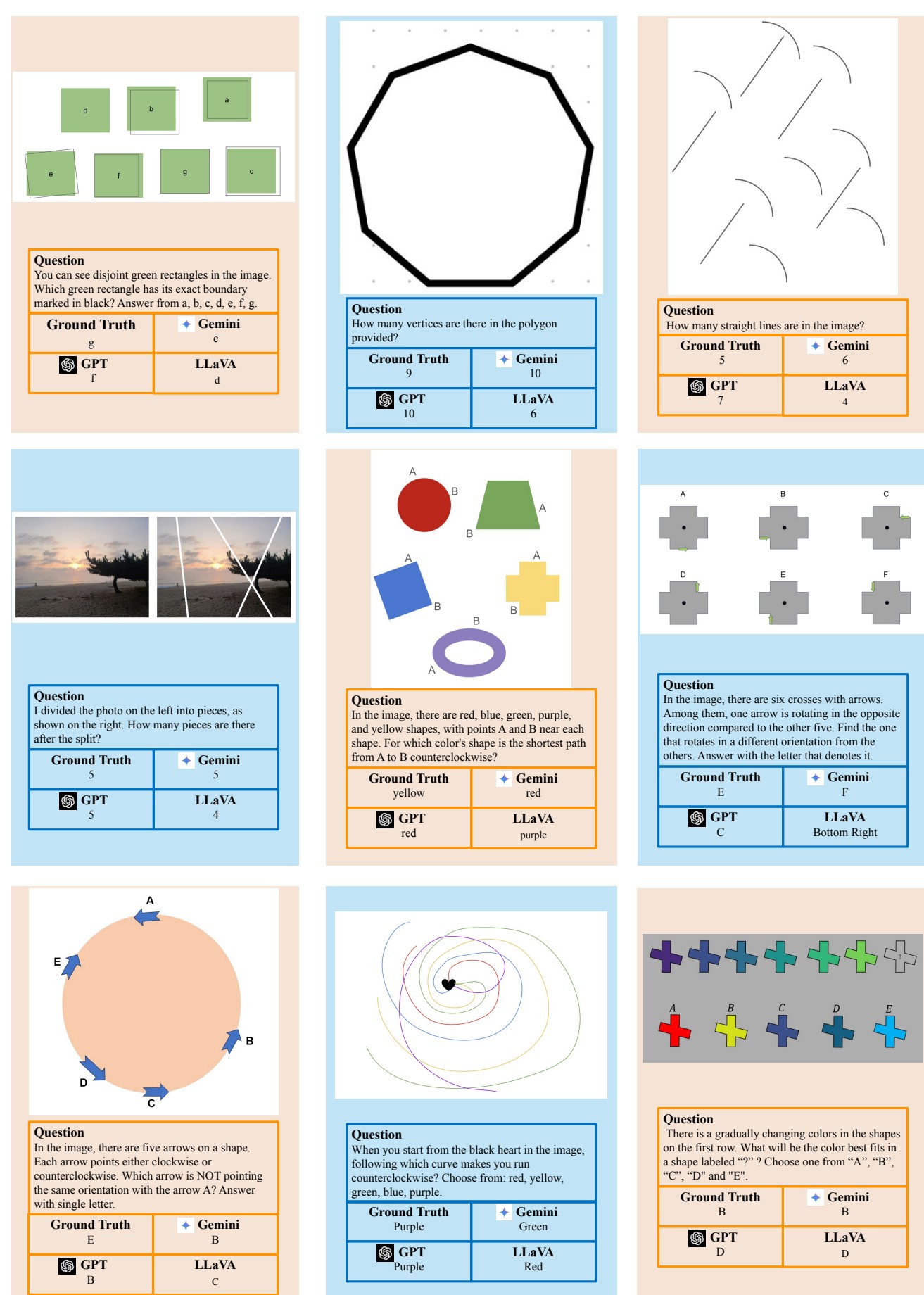

Figure 7: Sample problems from AVSBench and responses from Gemini-1.5-pro, GPT-4o, and LLaVA-Next-13B, 2/11.

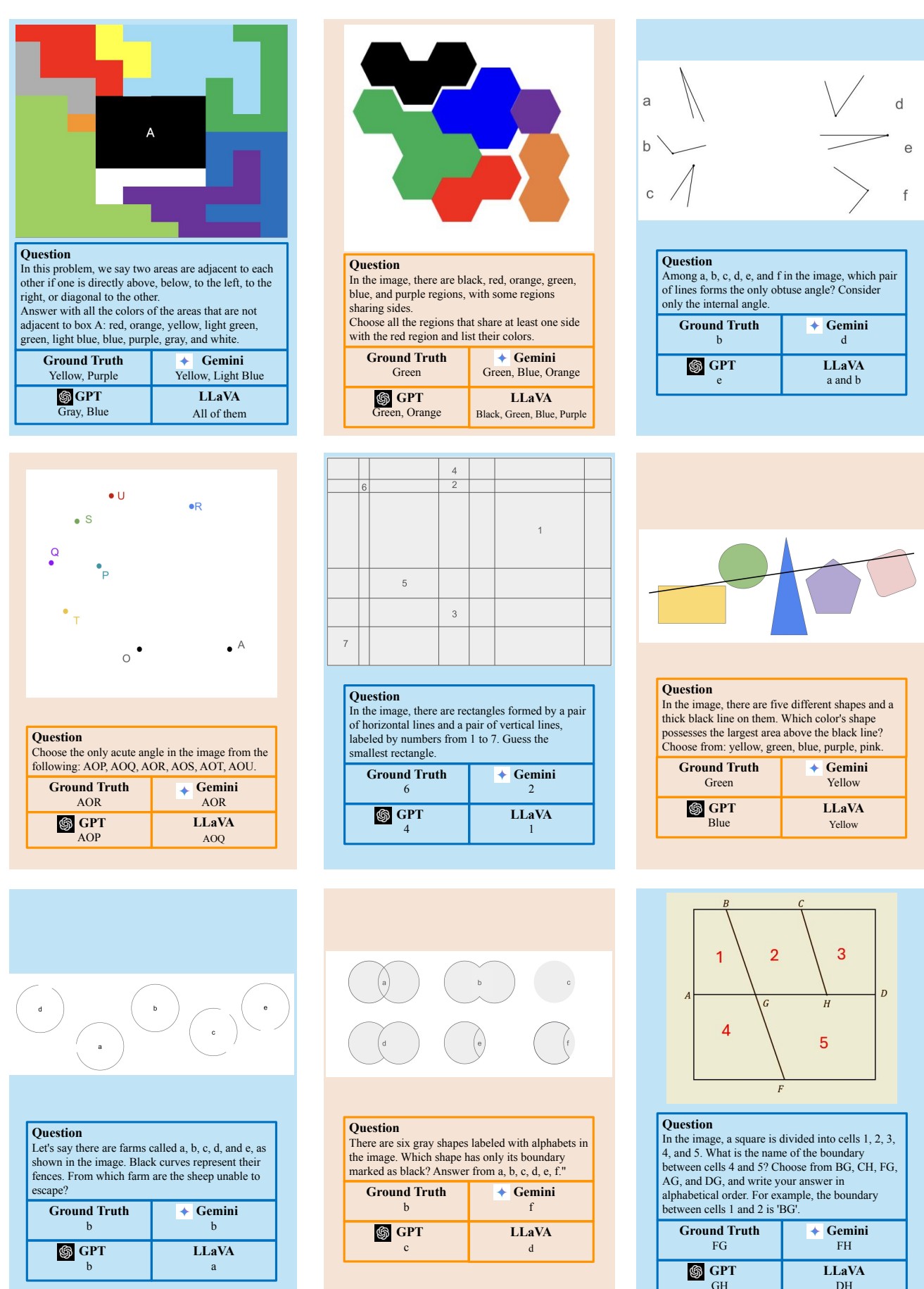

Figure 8: Sample problems from AVSBench and responses from Gemini-1.5-pro, GPT-4o, and LLaVA-Next-13B, 3/11.

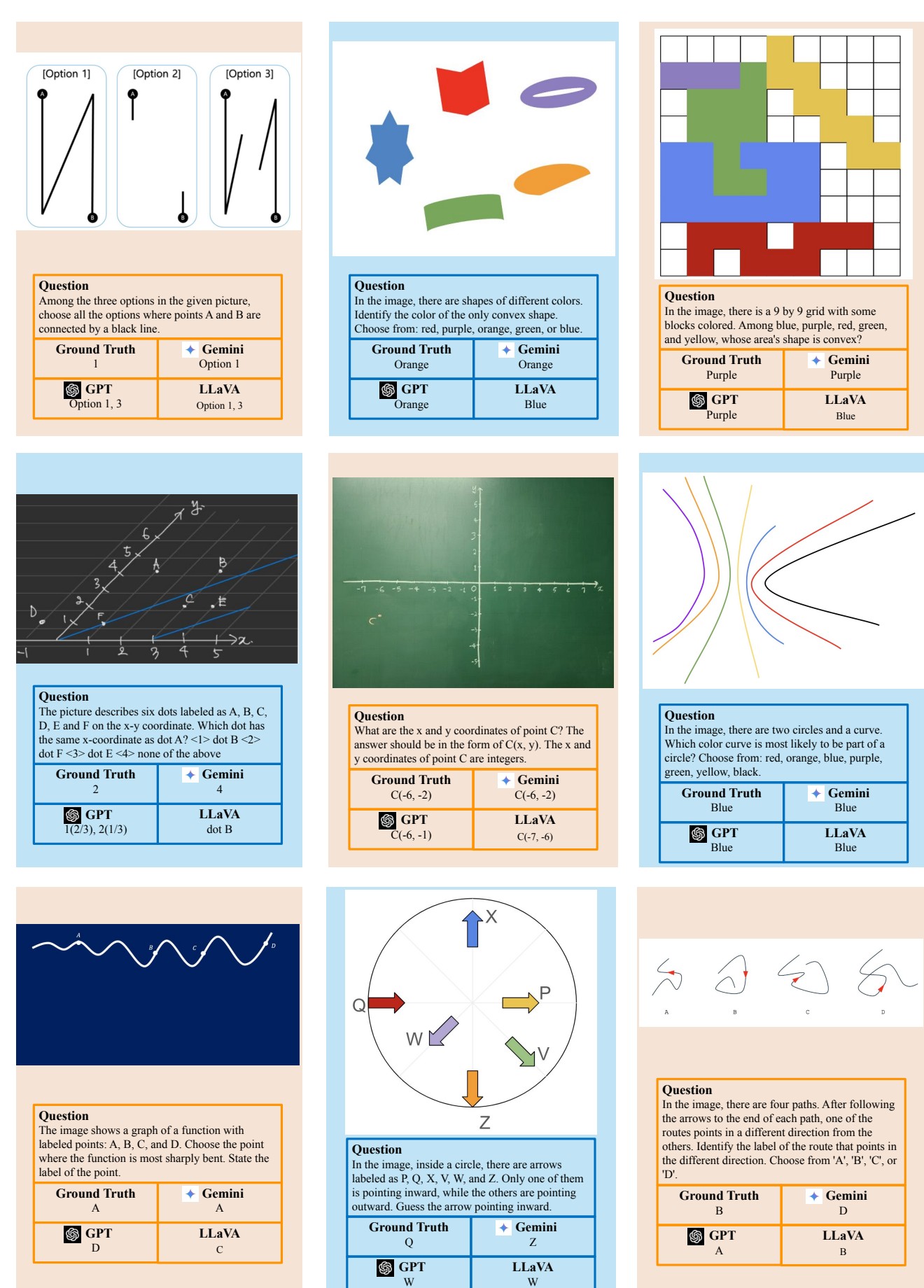

Figure 9: Sample problems from AVSBench and responses from Gemini-1.5-pro, GPT-4o, and LLaVA-Next-13B, 4/11.

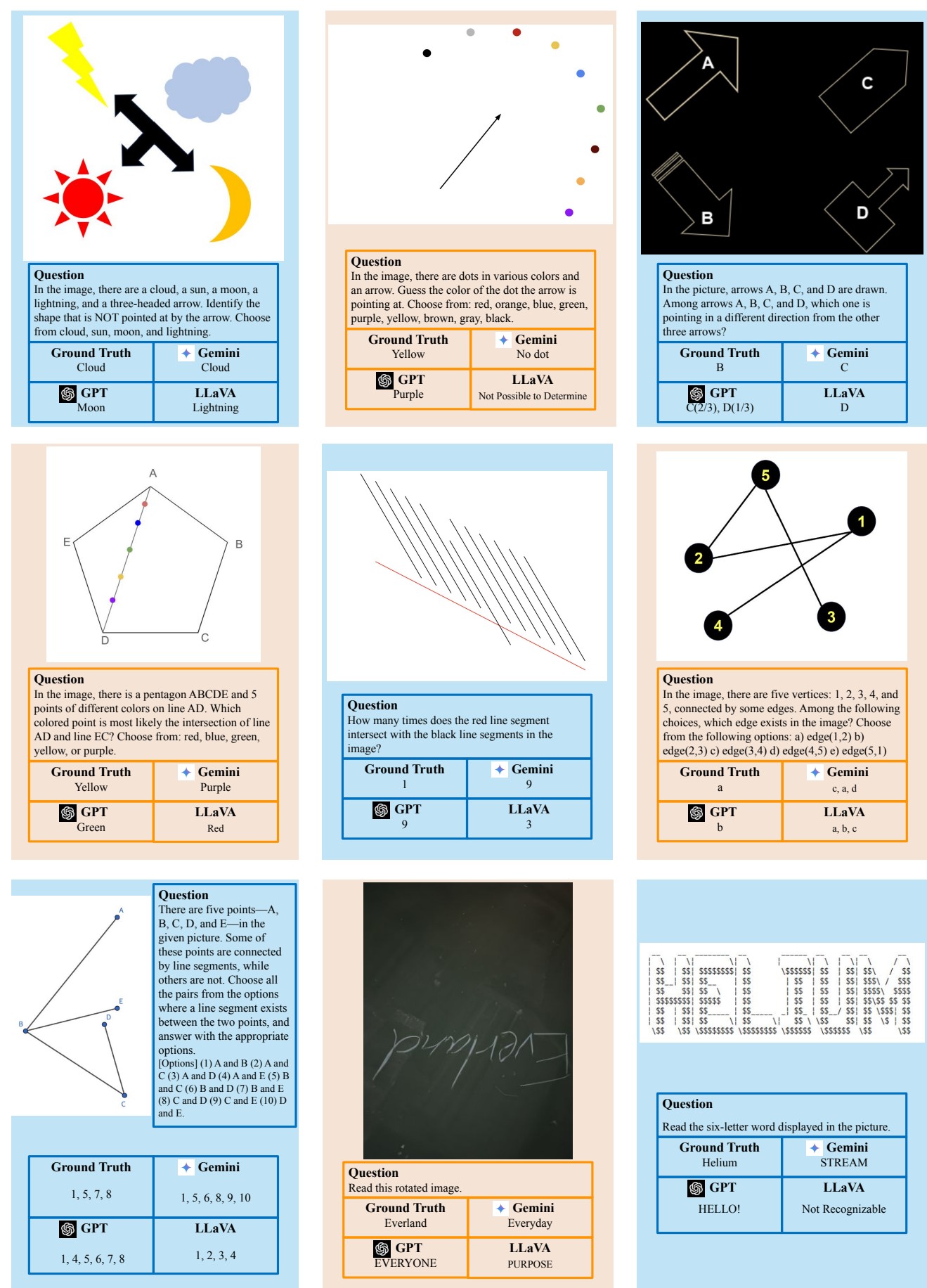

Figure 10: Sample problems from AVSBench and responses from Gemini-1.5-pro, GPT-4o, and LLaVA-Next-13B, 5/11.

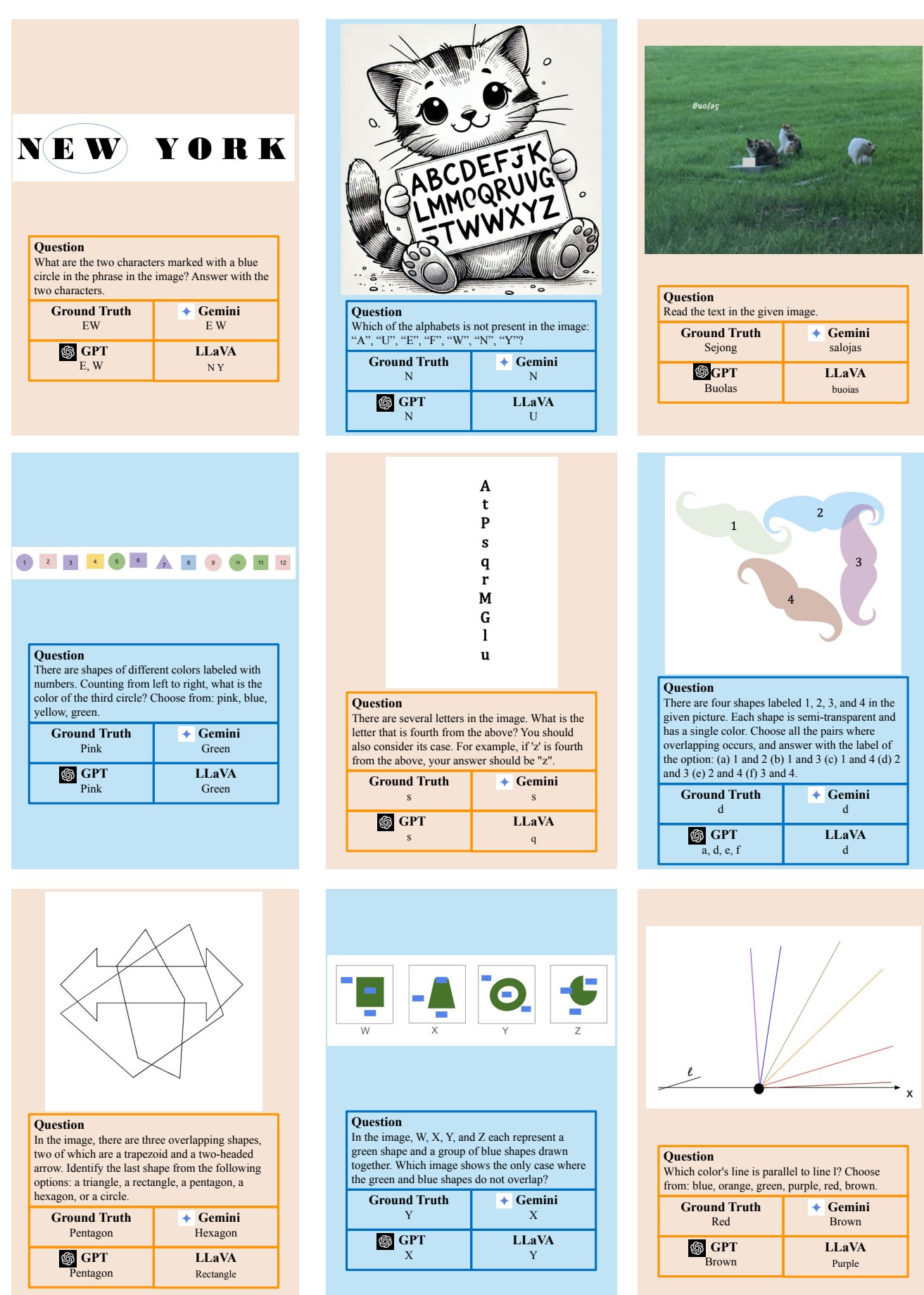

Figure 11: Sample problems from AVSBench and responses from Gemini-1.5-pro, GPT-4o, and LLaVA-Next-13B, 6/11.

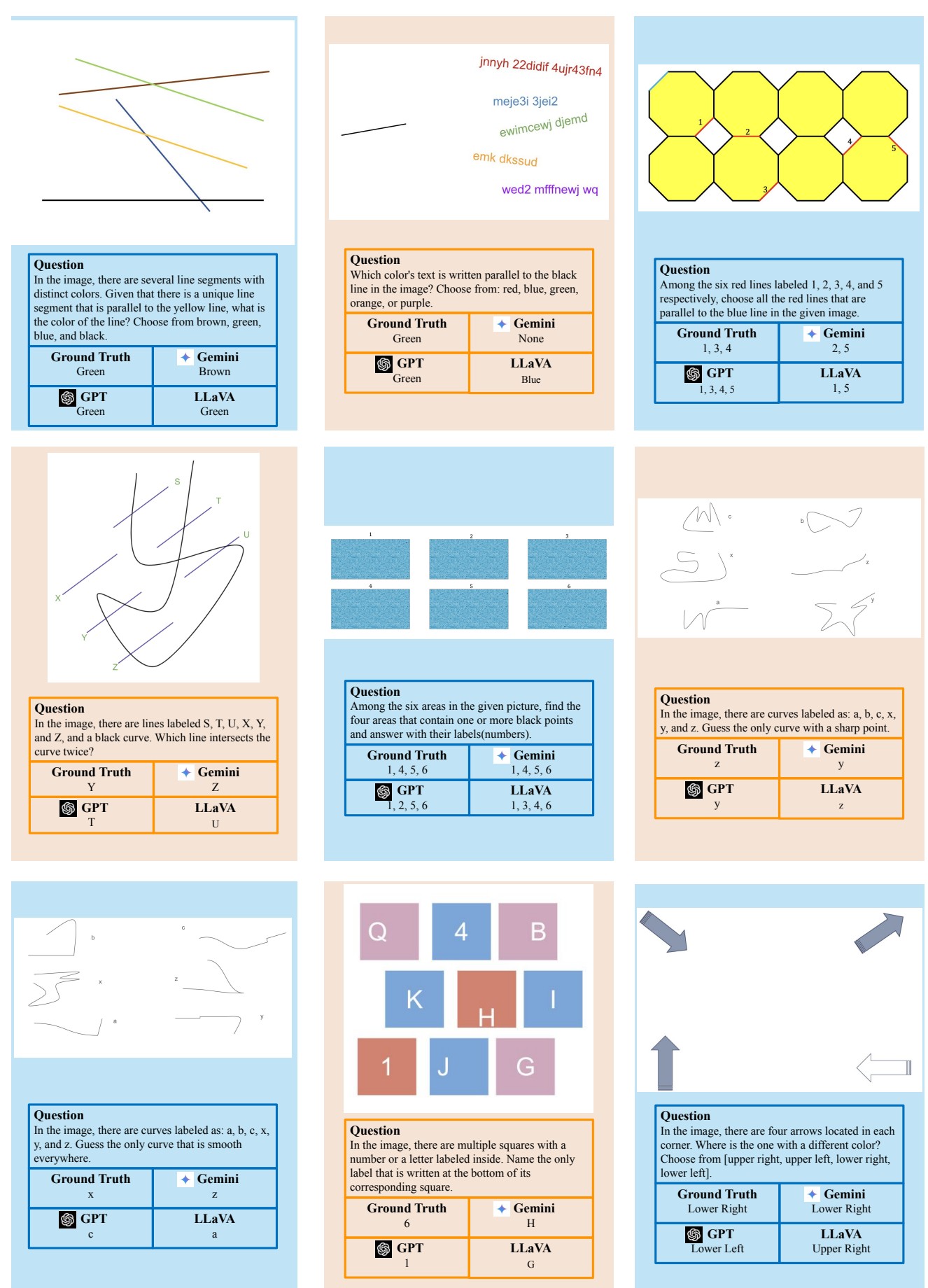

Figure 12: Sample problems from AVSBench and responses from Gemini-1.5-pro, GPT-4o, and LLaVA-Next-13B, 7/11.

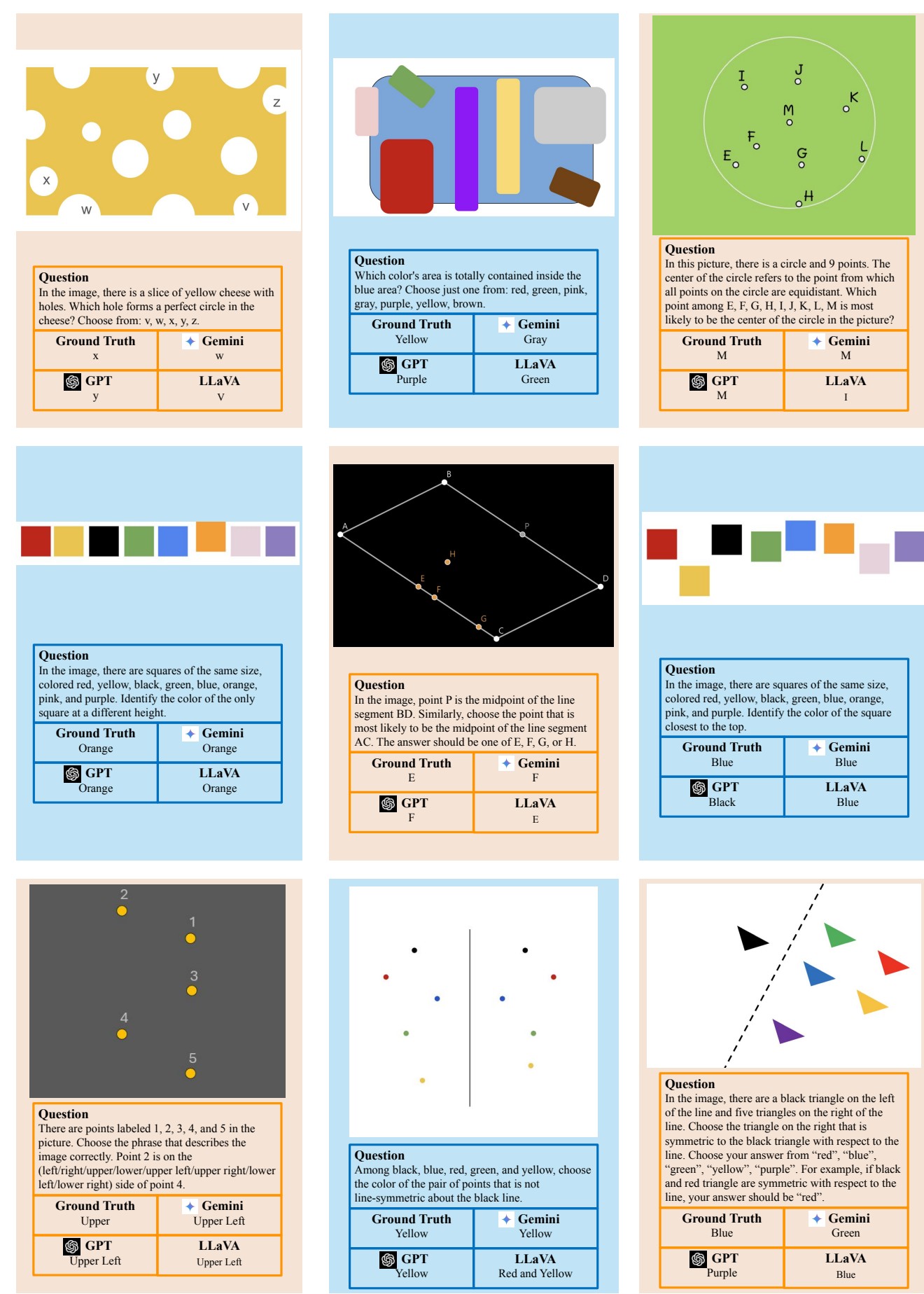

Figure 13: Sample problems from AVSBench and responses from Gemini-1.5-pro, GPT-4o, and LLaVA-Next-13B, 8/11.

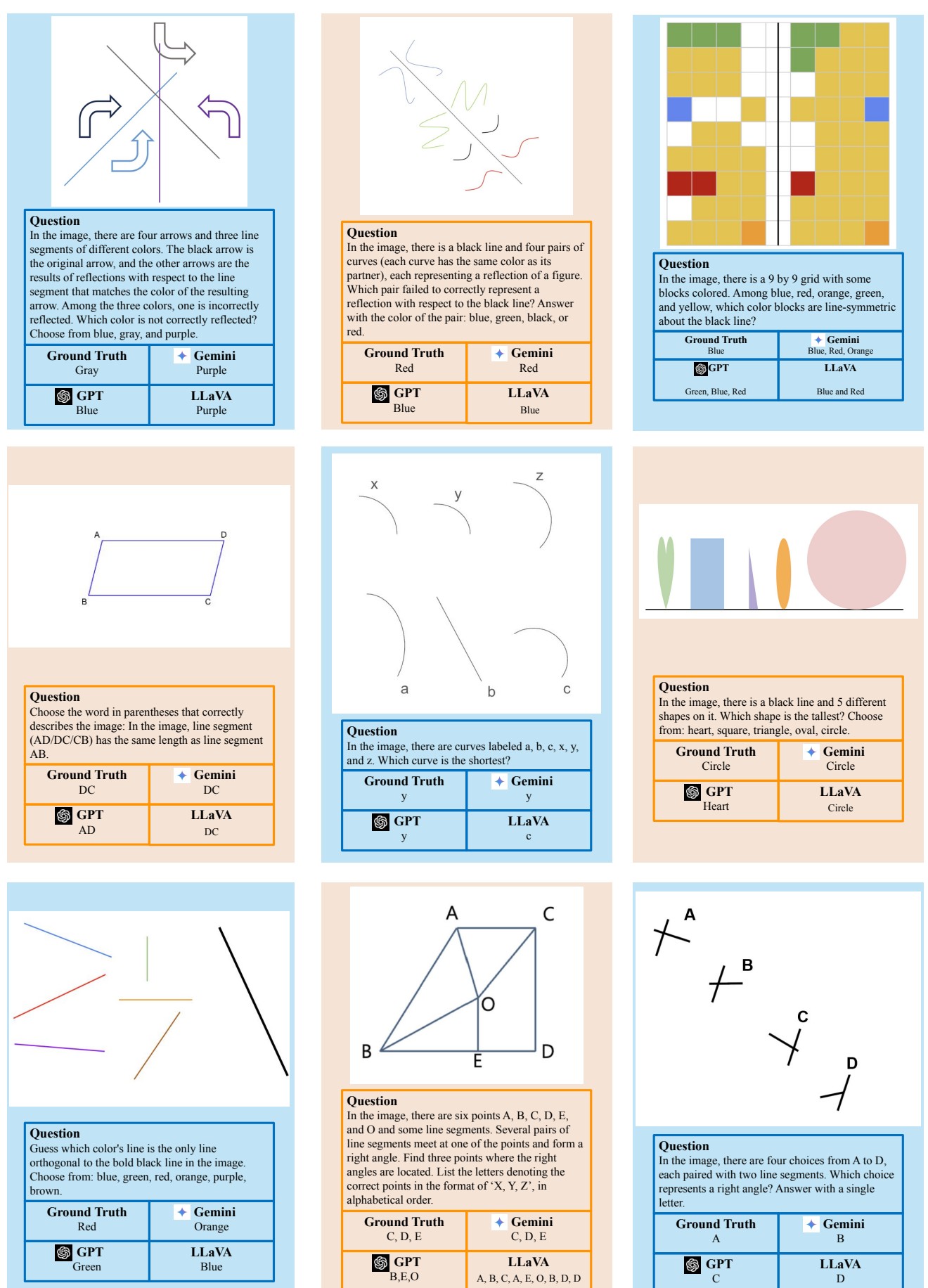

Figure 14: Sample problems from AVSBench and responses from Gemini-1.5-pro, GPT-4o, and LLaVA-Next-13B, 9/11.

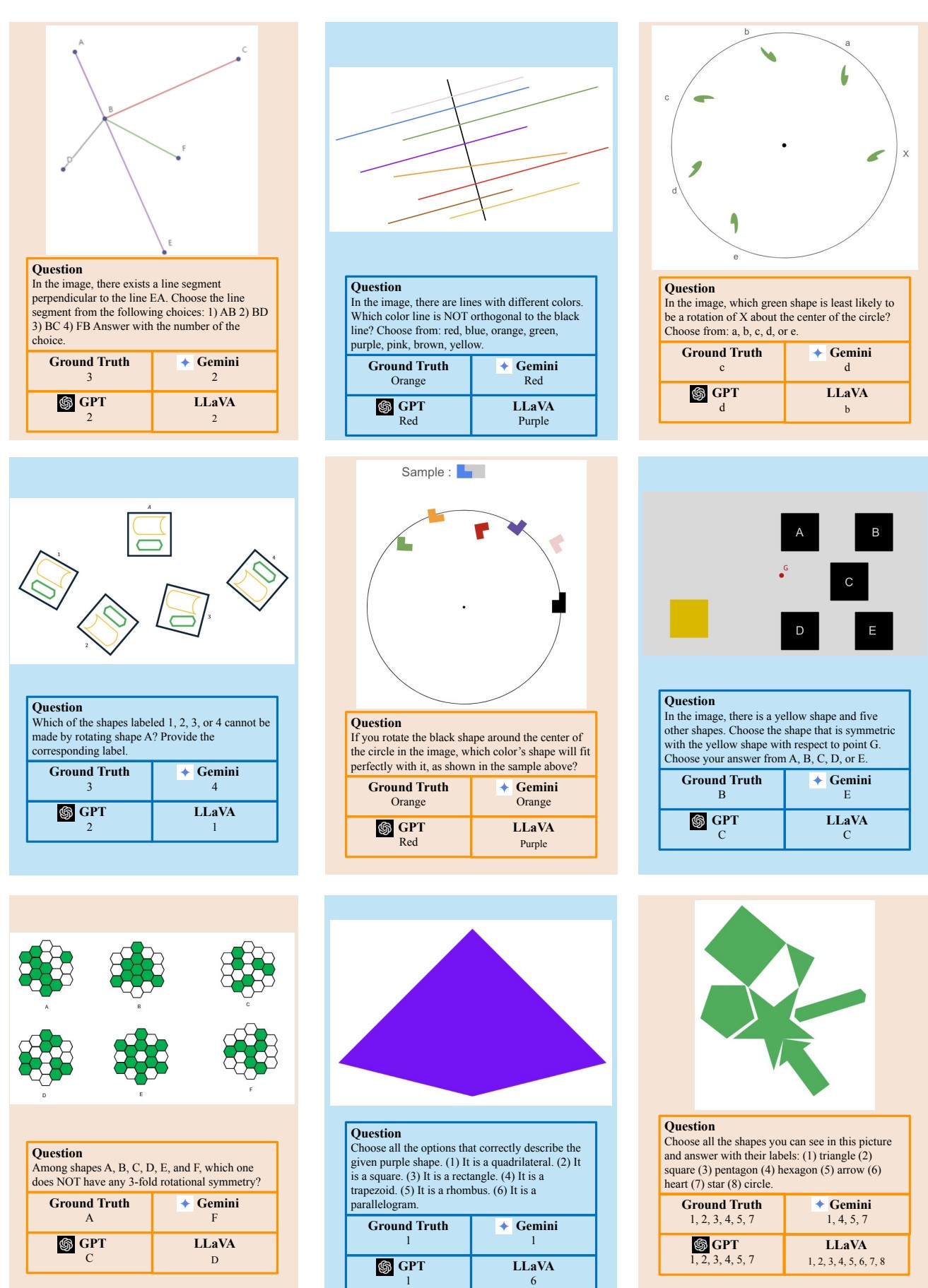

Figure 15: Sample problems from AVSBench and responses from Gemini-1.5-pro, GPT-4o, and LLaVA-Next-13B, 10/11.

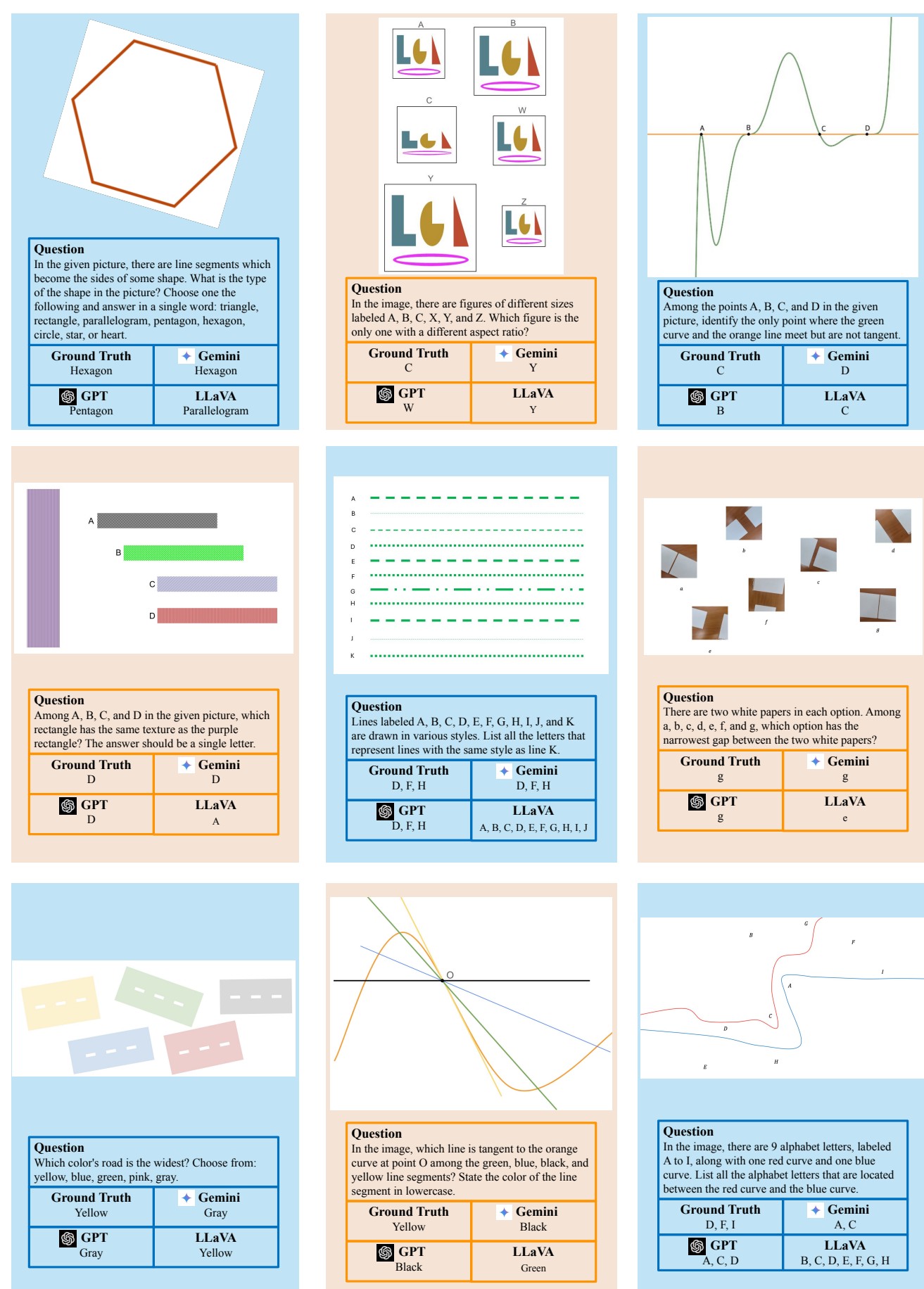

Figure 16: Sample problems from AVSBench and responses from Gemini-1.5-pro, GPT-4o, and LLaVA-Next-13B, 11/11.

# D    Descriptions of the atomic visual skills

In this section, we provide detailed definitions of the 36 atomic visual skills, together with a corresponding problem sample from AVSBench.

1. `Angle` is a skill to understand how an angle is visually represented. Angle is the primary factor in how a polygon looks, how two or more objects are related, and many other situations.

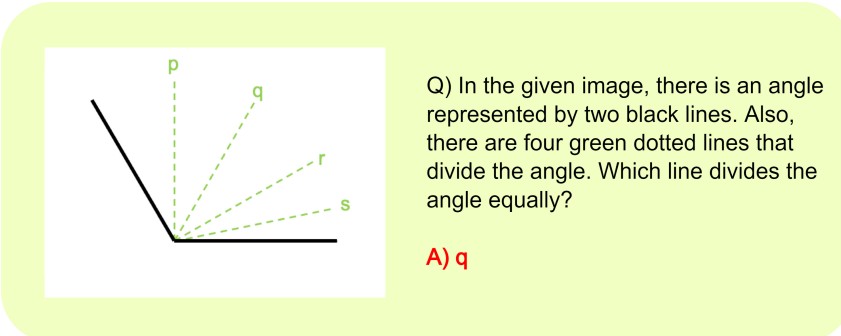

Q) In the given image, there is an angle represented by two black lines. Also, there are four green dotted lines that divide the angle. Which line divides the angle equally?

A) q

2. `Direction` is an ability to recognize linear direction in an image. It is a fundamental skill in human vision, supporting representation of linearity and multi-dimensional relations.

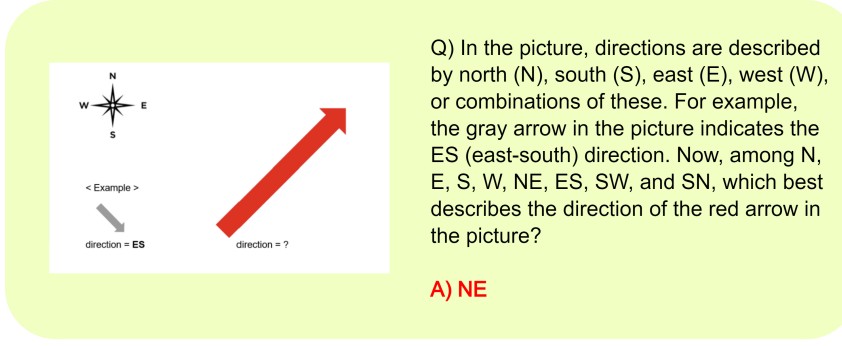

Q) In the picture, directions are described by north (N), south (S), east (E), west (W), or combinations of these. For example, the gray arrow in the picture indicates the ES (east-south) direction. Now, among N, E, S, W, NE, ES, SW, and SN, which best describes the direction of the red arrow in the picture?

A) NE

3. `Boundary` is a skill to understand the ends of objects or areas, and to detect visual representation of edges. The skill is used in distinguishing between distinct objects, or detecting boundaries between spaces.

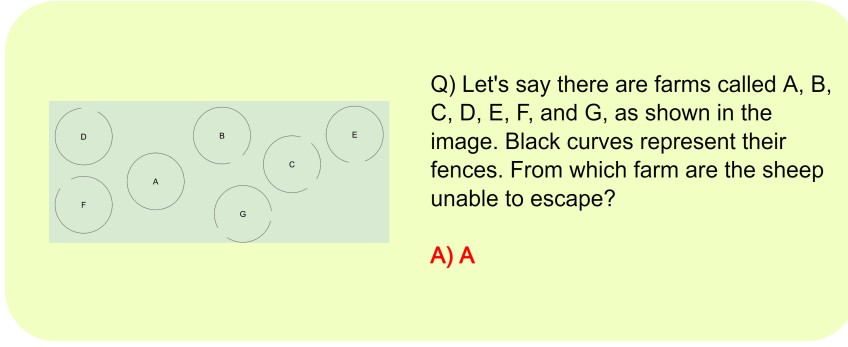

Q) Let's say there are farms called A, B, C, D, E, F, and G, as shown in the image. Black curves represent their fences. From which farm are the sheep unable to escape?

A) A

4. `Cardinal` is a field about counting distinct objects or specified concepts. Mastery of cardinals implies measuring quantities or dealing with multiple objects. Especially, it should take into account everything that satisfies given conditions, giving a difference from the skill of understanding *Ordinals*.

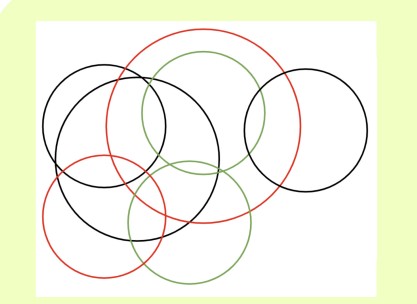

Q) How many circles are in the image?

A) 7

5. `Congruence` is a skill of detecting objects with the exact same scale and shape, and understanding their correspondence. Congruence is a primary component of visualizing various symmetries including translation, rotation or flipping. Congruence is distinguished from other equivalence because it requires the objects to be equal at all levels of measurement.

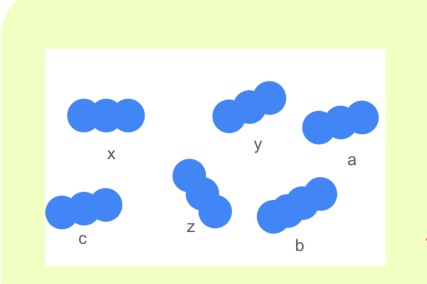

Q) In the image, there are different shapes labeled as a, b, c, x, y, and z. Guess which shape is different from the others.

A) b

6. `Convexity` is a skill of understanding convexity of given shapes. The skill is also closely related to detecting bumps or indentations and understanding convex and concave functions.

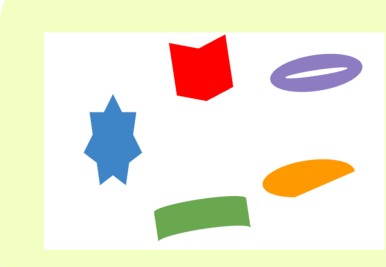

Q) In the image, there are shapes of different colors. Identify the color of the only convex shape. Choose from: red, purple, orange, green, or blue.

A) Orange

7. `Intersection` is a mastery of detecting intersections of lines and curves. The skill is necessary for interpreting relationships among 1-dimensional objects, and also among higher dimensional objects from 1-dimensional representations of their boundaries.

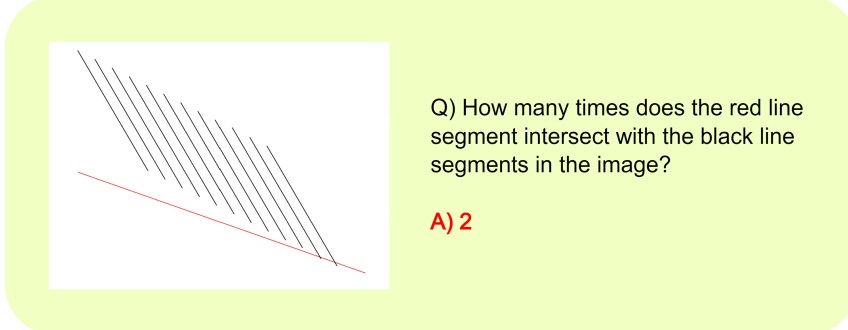

Q) How many times does the red line segment intersect with the black line segments in the image?

A) 2

8. `Line` is a skill to detect line segments and understand their roles in the image. This skill is a fundamental unit in understanding various objects as polygons, graphs and diagrams.

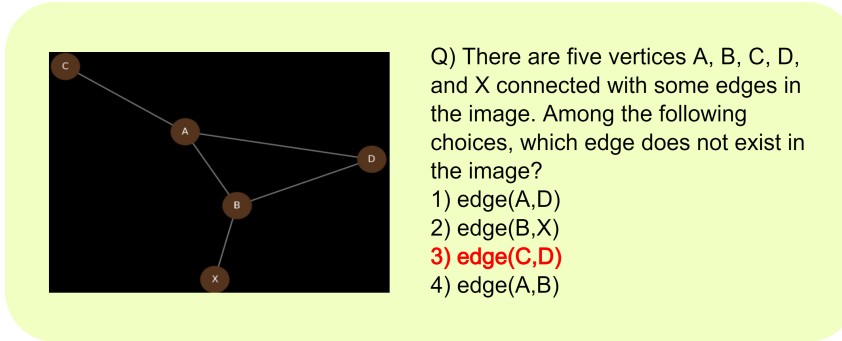

Q) There are five vertices A, B, C, D, and X connected with some edges in the image. Among the following choices, which edge does not exist in the image?
1) edge(A,D)
2) edge(B,X)
3) edge(C,D)
4) edge(A,B)

9. `OCR` is a skill to detect and read characters from visual inputs.

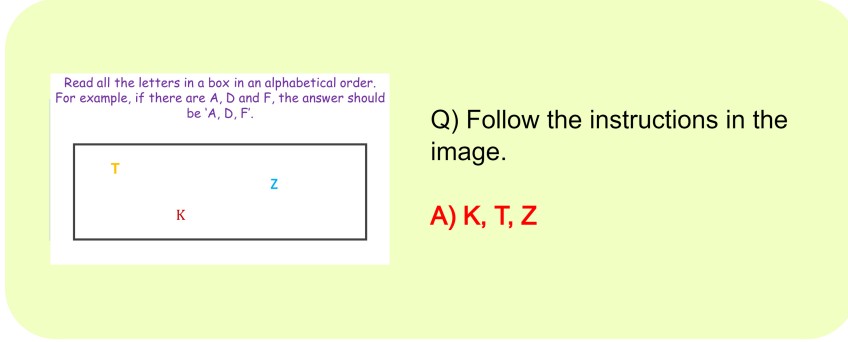

Q) Follow the instructions in the image.

A) K, T, Z

10. `Ordinal` is a skill to count certain objects or concepts in a given order. Mastery of this skill requires not just counting but also focusing on specific portions and order of targets, giving a difference from Cardinal Understanding.

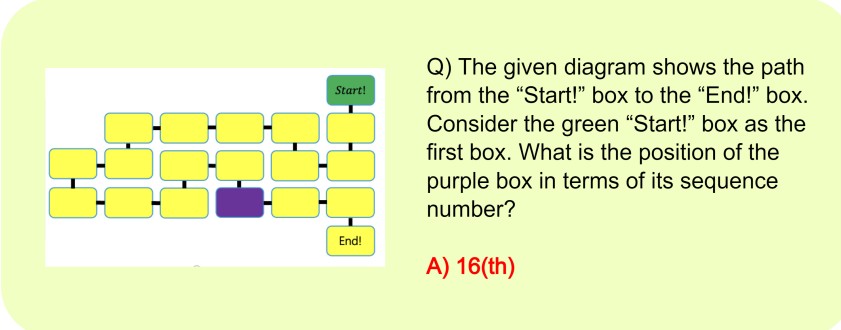

Q) The given diagram shows the path from the "Start!" box to the "End!" box. Consider the green "Start!" box as the first box. What is the position of the purple box in terms of its sequence number?

A) 16(th)

11. `Overlap` skill is about correctly recognizing two or more objects sharing a common area. The skill is crucial in understanding overlapping shapes or complex shapes such as diagrams.

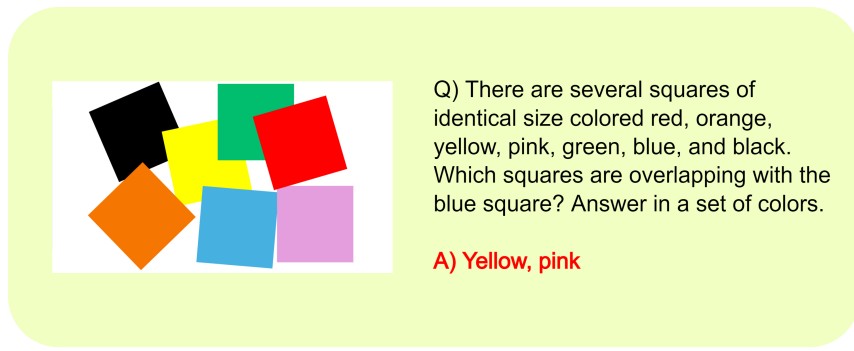

Q) There are several squares of identical size colored red, orange, yellow, pink, green, blue, and black. Which squares are overlapping with the blue square? Answer in a set of colors.

A) Yellow, pink

12. `Interior` is a skill of distinguishing between interior and exterior of the target area. This skill is essential in perceiving different areas.

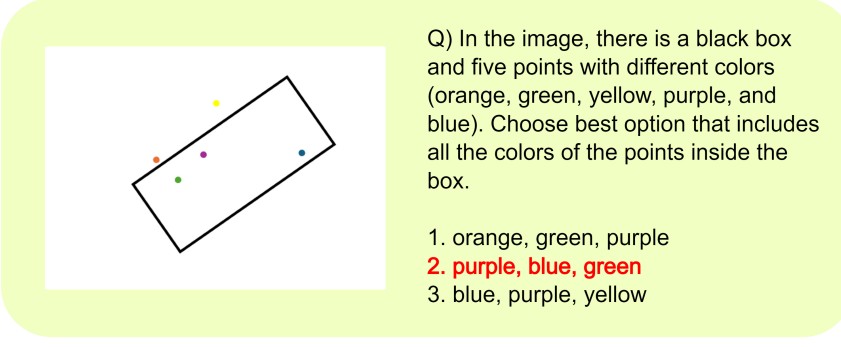

Q) In the image, there is a black box and five points with different colors (orange, green, yellow, purple, and blue). Choose best option that includes all the colors of the points inside the box.

1. orange, green, purple
2. purple, blue, green
3. blue, purple, yellow

13. `Relative Position` is an ability to identify positional relationships between objects that cannot be simply described such as inside, outside, or moved in a certain direction. This skill requires comprehension of complex relationships such as "positioned in between," or "at the same side of."

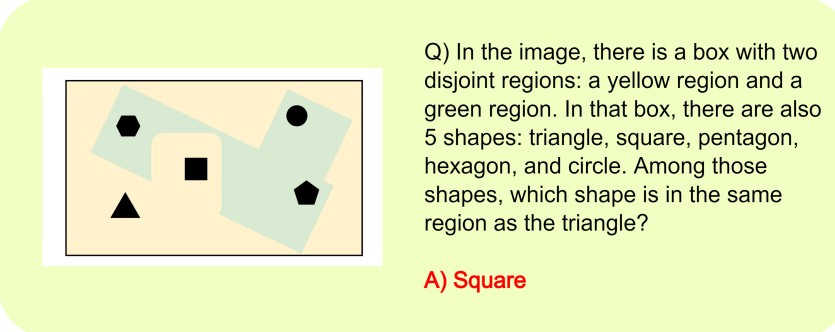

Q) In the image, there is a box with two disjoint regions: a yellow region and a green region. In that box, there are also 5 shapes: triangle, square, pentagon, hexagon, and circle. Among those shapes, which shape is in the same region as the triangle?

A) Square

14. `Reflection` is a field of recognizing linear symmetries. It requires detecting the axis of reflection and induced correspondence of objects.

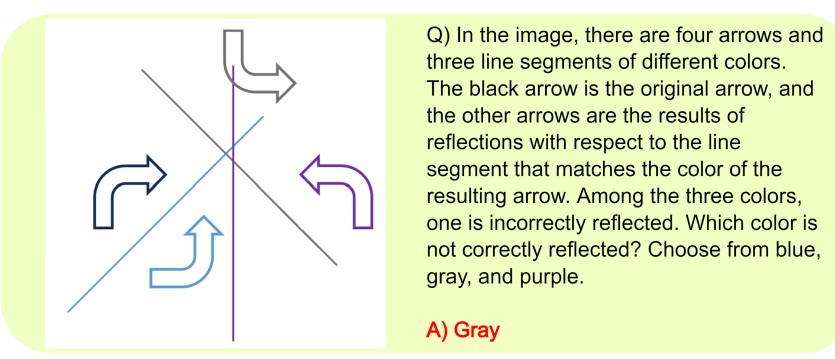

Q) In the image, there are four arrows and three line segments of different colors. The black arrow is the original arrow, and the other arrows are the results of reflections with respect to the line segment that matches the color of the resulting arrow. Among the three colors, one is incorrectly reflected. Which color is not correctly reflected? Choose from blue, gray, and purple.

A) Gray

15. `Length` is a skill to handle lengths of different objects. It involves comparing different lengths and measuring distances of objects.

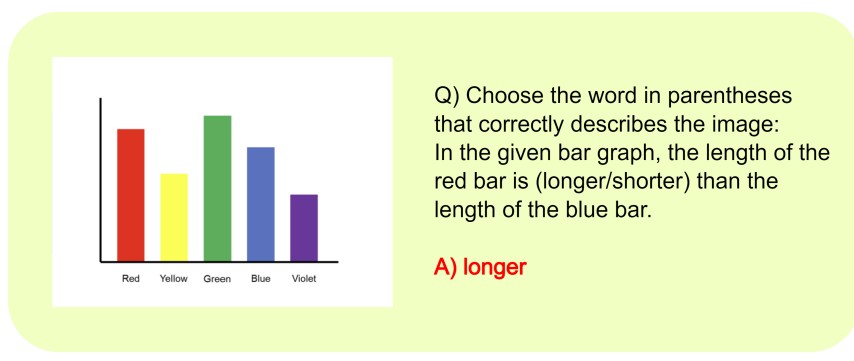

Q) Choose the word in parentheses that correctly describes the image: In the given bar graph, the length of the red bar is (longer/shorter) than the length of the blue bar.

A) longer

16. `Rotation` is an ability to identify changes in positions and angles induced by rotation, and detecting the axis of rotation.

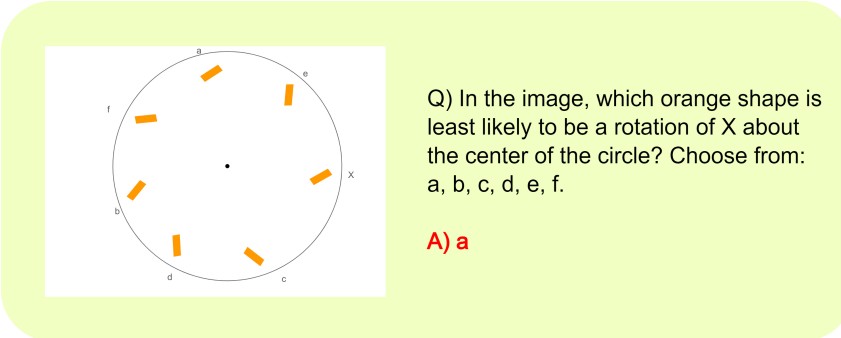

Q) In the image, which orange shape is least likely to be a rotation of X about the center of the circle? Choose from: a, b, c, d, e, f.

A) a

17. `Rotational Symmetry` is a field of symmetric representations with respect to rotations. The skill involves understanding invariant geometric features under specified rotations.

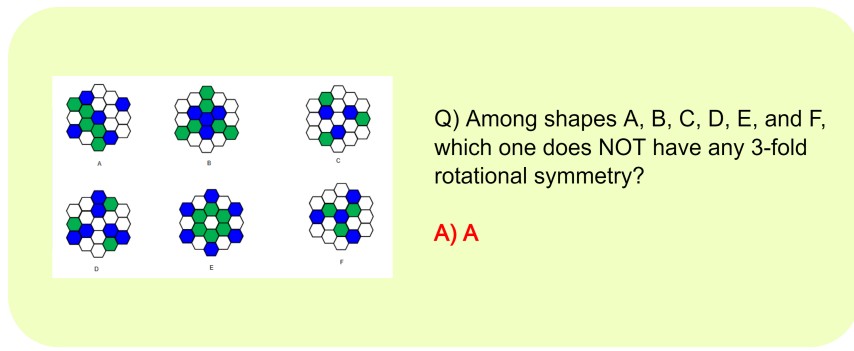

Q) Among shapes A, B, C, D, E, and F, which one does NOT have any 3-fold rotational symmetry?

A) A

18. `Symbol` is a skill to detect symbols, understand their roles in the image, and combine them with other visual information to attain the correct interpretation of the image.

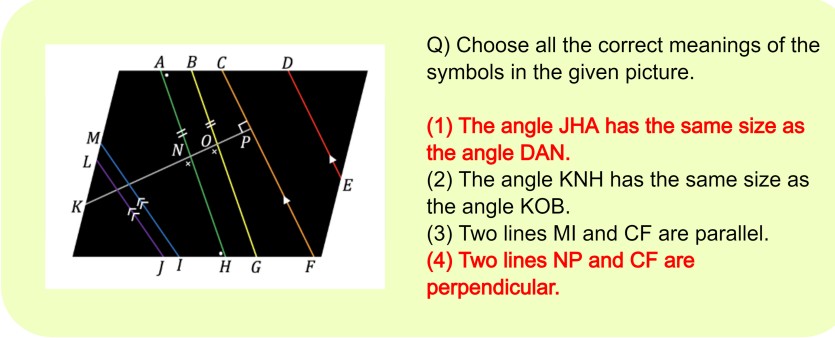

Q) Choose all the correct meanings of the symbols in the given picture.

(1) The angle JHA has the same size as the angle DAN.
(2) The angle KNH has the same size as the angle KOB.
(3) Two lines MI and CF are parallel.
(4) Two lines NP and CF are perpendicular.

19. `Texture` is a skill to understand textures of objects in the image. The skill is essential as texture is another main component of visual representation of objects, and is used to distinguish different objects with same shapes, such as line styles.

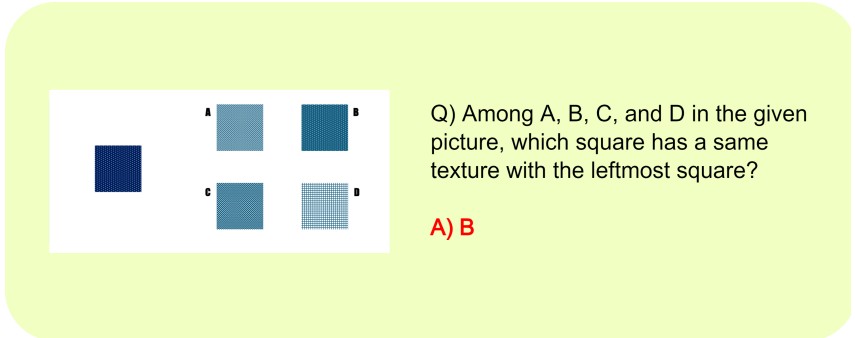

Q) Among A, B, C, and D in the given picture, which square has a same texture with the leftmost square?

A) B

20. `Width` is a skill to understand thickness and width of objects or areas. The skill is essential in measuring area or proportion of images together with length understanding.

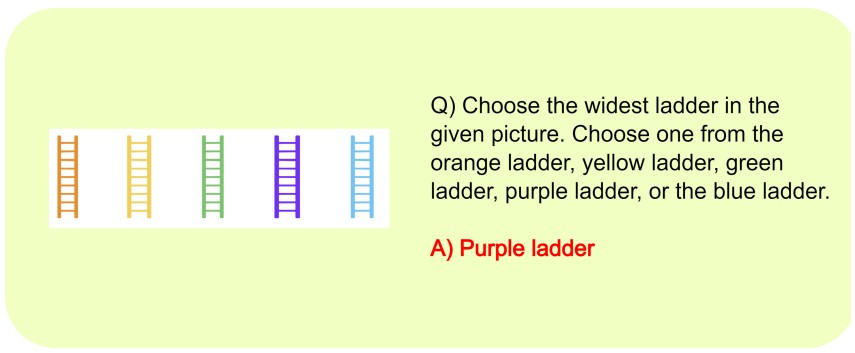

Q) Choose the widest ladder in the given picture. Choose one from the orange ladder, yellow ladder, green ladder, purple ladder, or the blue ladder.

A) Purple ladder

21. `Adjacency` is a skill to recognize when two or more objects are next to each other. The skill is crucial in understanding features induced by close positions such as forming clusters.

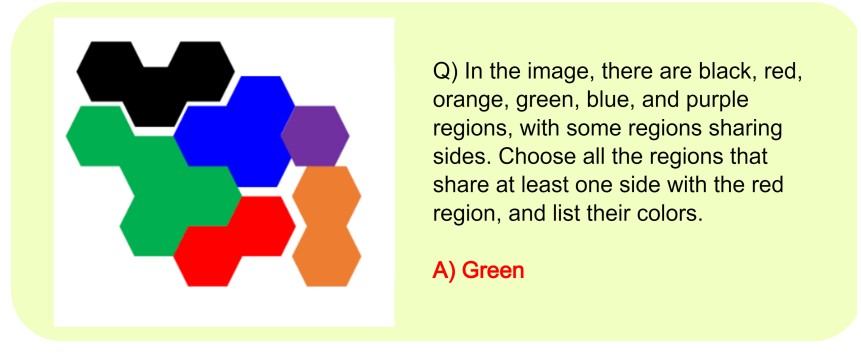

Q) In the image, there are black, red, orange, green, blue, and purple regions, with some regions sharing sides. Choose all the regions that share at least one side with the red region, and list their colors.

A) Green

22. `Absolute Position` is a skill to correctly understand where the objects are represented as a part of the visual input, independently of other objects. This involves recognizing objects posited at corners of an image, or comparing heights of objects represented in the image.

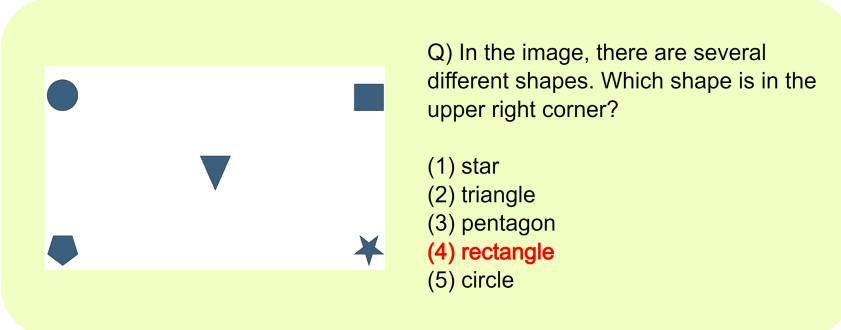

Q) In the image, there are several different shapes. Which shape is in the upper right corner?

(1) star
(2) triangle
(3) pentagon
(4) rectangle
(5) circle

23. `Area` is a skill to handle 2-dimensional volumes, including comparing areas.

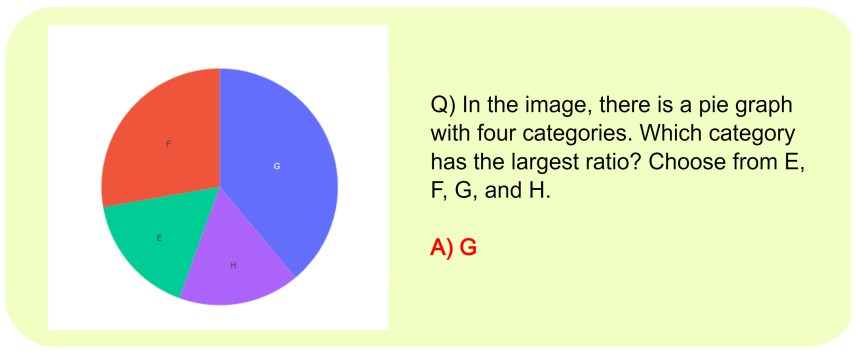

Q) In the image, there is a pie graph with four categories. Which category has the largest ratio? Choose from E, F, G, and H.

A) G

24. `Cardinal Direction` is a skill to understand primary directions including up, down, left, right, or diagonals. This involves recognizing North, South, West, and East directions.

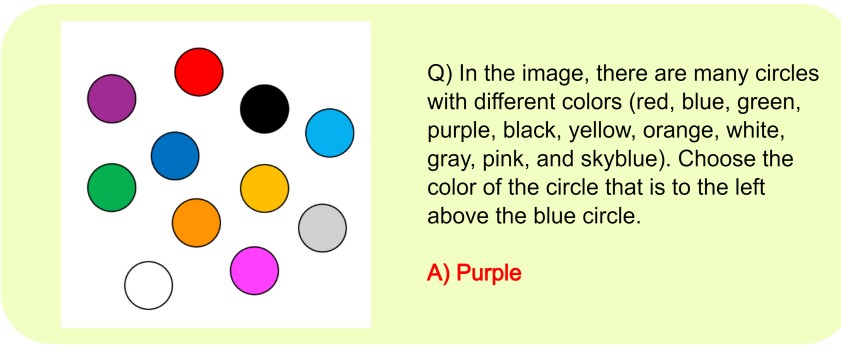

Q) In the image, there are many circles with different colors (red, blue, green, purple, black, yellow, orange, white, gray, pink, and skyblue). Choose the color of the circle that is to the left above the blue circle.

A) Purple

25. **Orthogonality** is a skill to identify orthogonal relations of objects in the image, including a right angle formed by two lines. Understanding orthogonality is fundamental in geometry, design, and engineering.

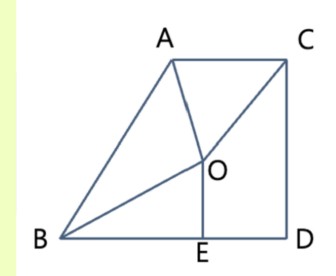

Q) In the image, there are six points A, B, C, D, E, and O and some line segments. Several pairs of line segments meet at one of the points and form a right angle. Find three points where the right angles are located.

A) C, D, E

26. **Tangency** is a skill to detect tangent objects. This skill focuses on geometric representation of tangent curves or boundaries, and is different from understanding adjacency that rather focuses on positional information.

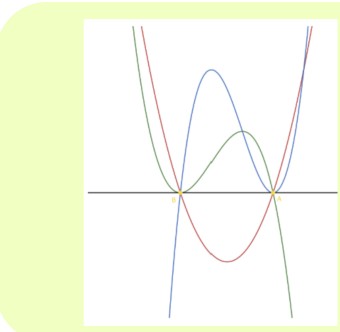

Q) In the given picture, there are three curves: red, blue, and green. Which curve is tangent to the black line at point B? State the color of that curve.

A) Green

27. **Connectedness** is a skill to identify connected components and detect links between objects. This is crucial in understanding interactions and distinguishing distinct components.

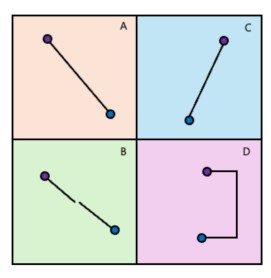

Q) Among option A, B, C, and D in the picture, choose one option where purple and blue points are not connected by a black line.

A) B

28. `Parallel` is a skill to recognize parallel lines or curves. This is essential in identifying fundamental objects like squares.

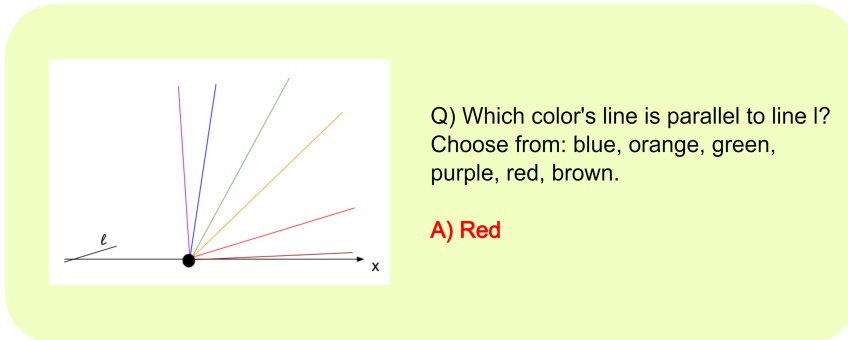

Q) Which color's line is parallel to line l? Choose from: blue, orange, green, purple, red, brown.

A) Red

29. `Similarity` is a skill to understand equivalence of geometric representations independent of scale. It also involves understanding of rescaling or comparing aspect ratios.

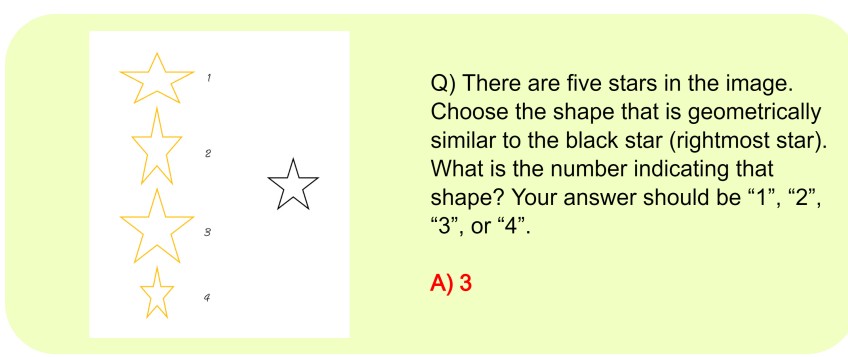

Q) There are five stars in the image. Choose the shape that is geometrically similar to the black star (rightmost star). What is the number indicating that shape? Your answer should be "1", "2", "3", or "4".

A) 3

30. `Color` is an ability to perceive, distinguish different colors, and understand the change in saturation and brightness.

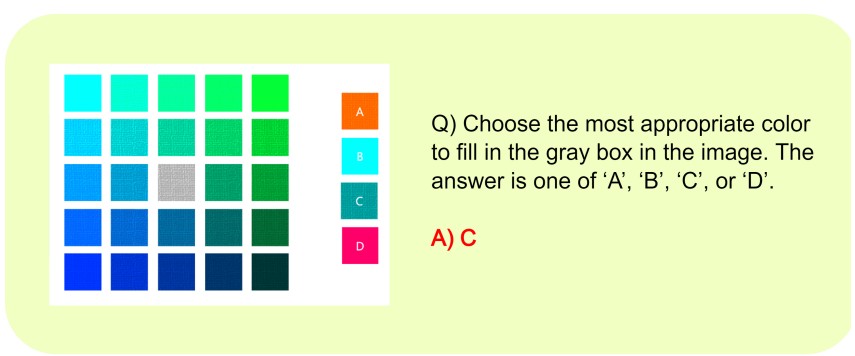

Q) Choose the most appropriate color to fill in the gray box in the image. The answer is one of 'A', 'B', 'C', or 'D'.

A) C

31. `Coordinate` is a skill to recognize and acquire correct information upon coordinate systems. We provide and acquire information about different systems such as polar coordinates.

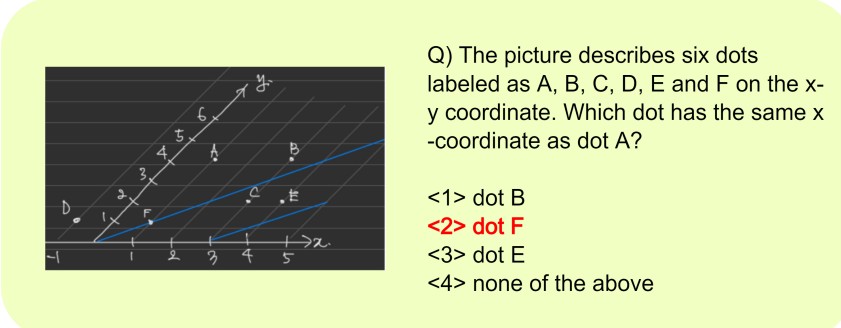

Q) The picture describes six dots labeled as A, B, C, D, E and F on the x-y coordinate. Which dot has the same x-coordinate as dot A?

<1> dot B
<2> dot F
<3> dot E
<4> none of the above

32. `Point` is a fundamental capability to detect points and understand their roles in the image. It also involves understanding nodes in different graphs.

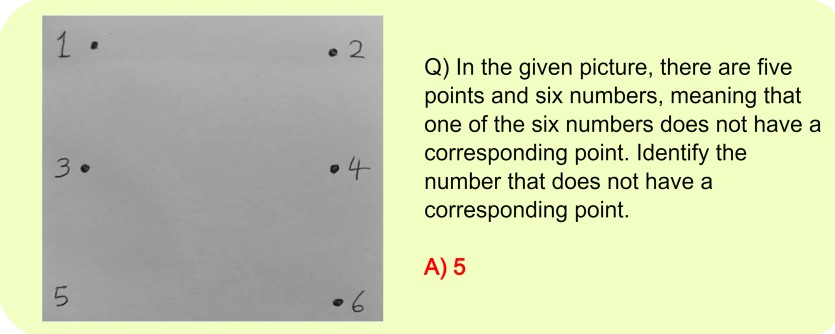

Q) In the given picture, there are five points and six numbers, meaning that one of the six numbers does not have a corresponding point. Identify the number that does not have a corresponding point.

A) 5

33. `Shape` is a skill to understand details of shapes and compare different shapes independently of positions or tilts. It also involves identifying popular shapes such as triangles, rectangles, circles, and stars.

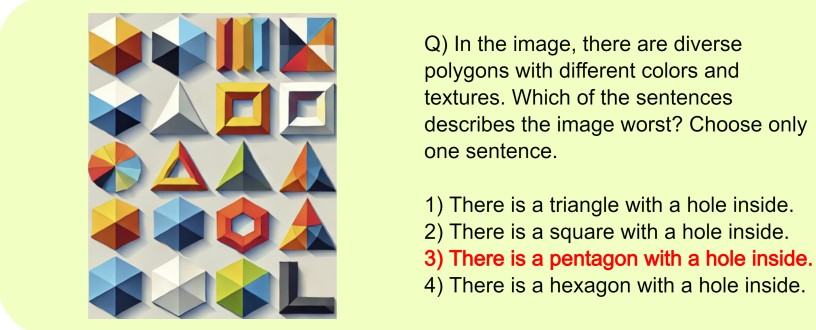

Q) In the image, there are diverse polygons with different colors and textures. Which of the sentences describes the image worst? Choose only one sentence.

1) There is a triangle with a hole inside.
2) There is a square with a hole inside.
3) There is a pentagon with a hole inside.
4) There is a hexagon with a hole inside.

34. `Curvature` is an ability to measure and compare curvatures of different curves. This involves distinguishing between straight lines and wavy curves, and detecting bends in a shape.

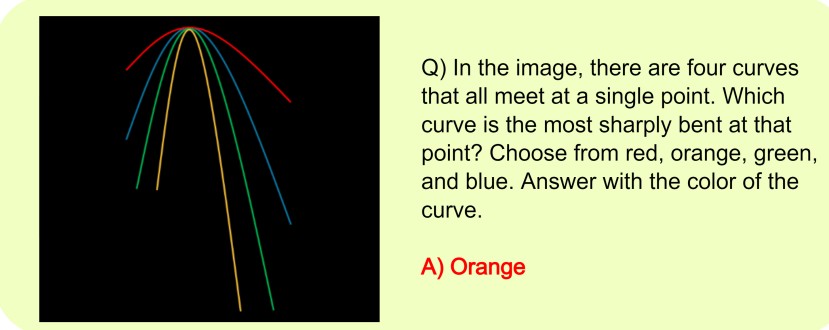

Q) In the image, there are four curves that all meet at a single point. Which curve is the most sharply bent at that point? Choose from red, orange, green, and blue. Answer with the color of the curve.

A) Orange

35. `Sharpness` is a skill to detect pointy parts of a shape. This is essential in understanding the representations of non-smooth objects such as points of a function that are not differentiable.

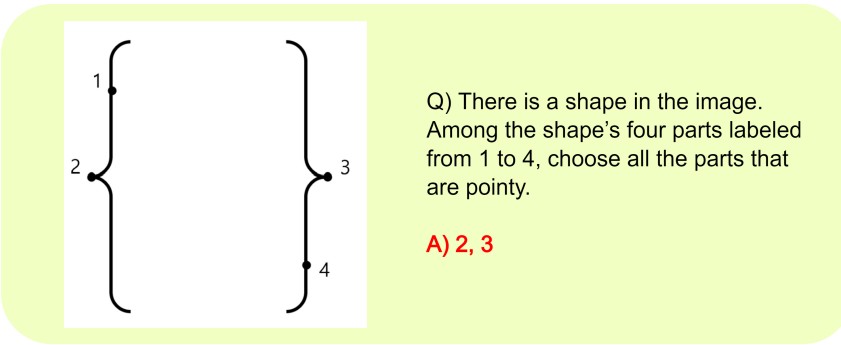

Q) There is a shape in the image. Among the shape's four parts labeled from 1 to 4, choose all the parts that are pointy.

A) 2, 3

36. `Orientation` is a skill to correctly distinguish clockwise and counterclockwise tendencies induced by not only rotations but also other movements that result in clockwise and counterclockwise directional change. The name originated from the mathematical definition of orientation in differential geometry.

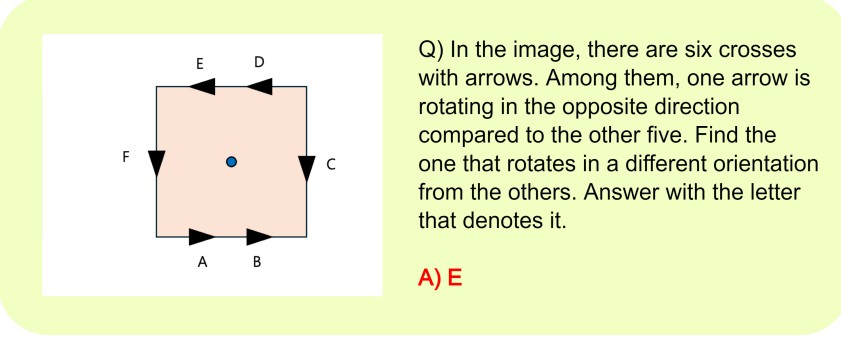

Q) In the image, there are six crosses with arrows. Among them, one arrow is rotating in the opposite direction compared to the other five. Find the one that rotates in a different orientation from the others. Answer with the letter that denotes it.

A) E

# E   Model versions

We evaluated closed-source models ChatGPT [40], Gemini [49] and open-weight models LLaVA-NeXT [29], LLaVA-OneVision [26], Math-LLaVA [47], Table-LLaVA [60], Phi-3.5-Vision [1], InternVL2 [10], DeepSeek-VL [31]. Tables 1 and 2 describe further details about the model sizes and versions. For closed-source models, we used the commercial APIs. All models' temperatures were set to 0.

Table 1: Versions of closed-source models

| Model Name | Version |
|---|---|
| ChatGPT | gpt-4o-2024-05-13 |
| | gpt-4o-mini-2024-07-18 (For scoring) |
| Gemini | gemini-1.5-pro-001 |

Table 2: Versions and model sizes of open-weight models

| Version | Model Size(s) |
|---|---|
| LLaVA-NeXT | 7B, 13B |
| LLaVA-OneVision | 7B |
| Math-LLaVA | 13B |
| Table-LLaVA | 7B |
| Phi-3.5-Vision-Instruct | 4B |
| InternVL2 | 8B |
| DeepSeek-VL | 7B |

# F   Further details on evaluation process

We used GPT-4o mini to extract answers from model responses and to judge correctness. Few-shot in-context learning prompts we provided to GPT-4o mini as described in Tables 3 and 4. To verify the reliability of this pipeline, we randomly selected 128 problems from our dataset and compared the scores from GPT-4o mini with human annotations. Reassuringly, GPT-4o mini and the human annotators agreed on the scoring of the 128 problems. We attribute this high level of reliability, in part, to the straightforward and clear design of our questions and answers.

| Element | Prompt |
|---|---|
| System prompt | Imagine you are an intelligent teacher. Thoroughly read the provided instruction to ensure a solid understanding of the information provided. |
| Task description | Please read the following example. Then extract the answer from the model response and type it at the end of the prompt. If the question requires a full sentence with a correct word filled in, please provide the word only.
{*examples*}
Question: {*question*}
Model response: {*model response*}
Extracted Answer: |
| Examples | **Question:** There is a single rectangle with multiple color layers in the image. What is the color of the boundary of the rectangle? The answer should be one of 'red', 'yellow', 'green', or 'blue'.
**Model response:** The color of the boundary of the circle is red.
**Extracted answer:** red

**Question:** How many line segments are in the image? Answer should be a number.
**Model response:** There are 4 dashed line segments in the image.
**Extracted answer:** 4

**Question:** Choose the word in parentheses that correctly describes the image. Rewrite the sentence with the chosen word.
In the image, shape (A/B) has sides curved inward. (Unit: $)
**Model response:** In the image, shape B has sides curved inward.
**Extracted answer:** B

**Question:** Choose the phrase in parentheses that correctly describes the image. Rewrite the sentence with the chosen phrase.
In the given image, the green arrow (is longer than/has the same length as/is shorter than) the black arrow.
**Model response:** In the given image, the green arrow is longer than the black arrow.
**Extracted answer:** is longer than

**Question:** In this image, choose the path which is a single line segment between points A and B from the following options. Provide your answer as a single uppercase letter: (A) the purple path (B) the blue path (C) the green path (D) the red path
**Model response:** B
**Extracted answer:** B

**Question:** Choose the most appropriate color to fill in the box marked with '?' in the image. The answer is one of 'a', 'b', 'c', or 'd'.
**Model response:** The correct color to fill in the box marked with '?' is (a) blue. The colors are following a gradient pattern from red, to a more purple hue, and finally to blue. The logical next color in the sequence would be blue, as it extends the progression seen in the previous squares.
**Extracted answer:** a

**Question:** There is a book in the image. What is the color of the book in the image? Choose answer from the number of the option and give your answer in "1", "2", "3", or "4". (1) red (2) yellow (3) blue (4) green
**Model response:** The color of the guitar in the image is (2) yellow.
**Extracted answer:** 2 |

Table 3: System prompt, task description, and examples used to prompt GPT-4o mini for answer extraction.

| Element | Prompt |
| --- | --- |
| System prompt | Imagine you are an intelligent teacher. Thoroughly read the provided instruction to ensure a solid understanding of the information provided. |
| Task description | The [Standard Answer] is the correct answer to the question, and the [Model Answer] is the answer generated by a model for that question. Thoroughly read both the [Standard Answer] and the [Model Answer]. Assess the consistency of the information provided in these two responses. |
| | Although you do not know the specific question, you can still assess the consistency between the two responses by checking for logical conflicts if both responses are assumed to be correct. |
| | If the [Model Answer] is consistent with the [Standard Answer], please answer '1'. Otherwise, answer '0'. |
| | When the [Standard Answer] is provided as a list, answer '1' if the [Model Answer] is consistent with at least one item on the list. Otherwise, answer '0'. |
| | Below are the examples of the correct consistency judgment. |
| | {*examples*} |
| | Now, below are two answers to a question. What is your judgment? |
| | [Standard Answer] {*standard answer*} |
| | [Model Answer] {*extracted answer*} |
| | Judgment: |
| Examples | **[Standard Answer]** a |
| | **[Model Answer]** a |
| | **Judgment:** 1 |
| | |
| | **[Standard Answer]** 1 |
| | **[Model Answer]** 4 |
| | **Judgment:** 0 |
| | |
| | **[Standard Answer]** circle |
| | **[Model Answer]** the circle |
| | **Judgment:** 1 |
| | |
| | **[Standard Answer]** 4 |
| | **[Model Answer]** shape 4 |
| | **Judgment:** 1 |
| | |
| | **[Standard Answer]** line segment B and C |
| | **[Model Answer]** B, C |
| | **Judgment:** 1 |
| | |
| | **[Standard Answer]** ac |
| | **[Model Answer]** ca |
| | **Judgment:** 0 |
| | |
| | **[Standard Answer]** 2 |
| | **[Model Answer]** two |
| | **Judgment:** 1 |
| | |
| | **[Standard Answer]** three |
| | **[Model Answer]** 3 |
| | **Judgment:** 1 |
| | |
| | **[Standard Answer]** ['ac', 'ca'] |
| | **[Model Answer]** ca |
| | **Judgment:** 1 |

Table 4: System prompt, task description, and examples used to prompt GPT-4o mini for judgment.

# G    Results on 36 skills and Further Analysis

In this section, we provide full details of Section 3 and further analysis of Findings 3.

Table 5 presents our full evaluation results on AVSBench. "Random Chance" represents the expected accuracy of randomly choosing multiple-choice questions and scoring 0 from all non-multiple-choice questions. The subcolumn named "overall" indicates the accuracy across all problems of its corresponding skill. The column named "**TOTAL**" describes the overall accuracy across all problems in AVSBench, involving all skills.

As mentioned in Findings 3, CoT did not provide meaningful performance gains when used with GPT-4o and Gemini-1.5-pro. Gemini-1.5-pro had only a $0.1\%$ gain from applying CoT, and GPT-4o had a $1\%$ gain. However, CoT worsened performances of skills including OCR, Length, and Symbol. As in the case of Figure 17, by inspecting the responses of GPT-4o with and without CoT prompting, we observe that the additional reasoning steps afforded by CoT are not helpful in comprehending visual inputs.

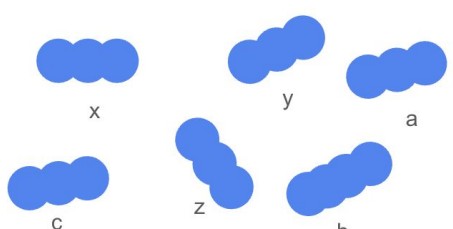

**Question**
In the image, there are different shapes labeled as a, b, c, x, y, and z. Guess which shape is different from the others.

**Ground Truth**
b

**GPT-4o**
The shape labeled \"z\" is different from the others. While all the other shapes (a, b, c, x, y) are arranged horizontally, the shape labeled \"z\" is arranged vertically.

**GPT-4o + CoT**
Let's analyze the shapes step by step:\n\n1. **Shape x**: Consists of three circles in a horizontal line.\n2. **Shape y**: Consists of three circles in a horizontal line.\n3. **Shape a**: Consists of three circles in a horizontal line.\n4. **Shape c**: Consists of three circles in a horizontal line.\n5. **Shape b**: Consists of three circles in a horizontal line.\n6. **Shape z**: Consists of three circles in a vertical line.\n\nAll shapes except for shape z consist of three circles in a horizontal line. Shape z is the only one with circles arranged vertically.\n\nTherefore, the shape that is different from the others is **shape z**.

Figure 17: Responses of GPT-4o on a problem from AVSBench with and without chain-of-thought (CoT) prompt.

Table 5: Full details of evaluation results on AVSBench. Each value represents the accuracy of the model of its row, on problems with the difficulty and for the skill of its column.

**Skills block 1: TOTAL, OCR, Absolute position, Adjacency, Angle, Area, Boundary**

| | TOTAL | | | | OCR | | | | Absolute position | | | | Adjacency | | | | Angle | | | | Area | | | | Boundary | | | |
|---|---|---|---|---|---|---|---|---|---|---|---|---|---|---|---|---|---|---|---|---|---|---|---|---|---|---|---|---|
| | easy | medium | hard | overall | easy | medium | hard | overall | easy | medium | hard | overall | easy | medium | hard | overall | easy | medium | hard | overall | easy | medium | hard | overall | easy | medium | hard | overall |
| Random Chance | 22.4 | 19.1 | 11.7 | 19.2 | 4.1 | 6.9 | 0.8 | 4.6 | 20.8 | 19.0 | 12.5 | 18.2 | 4.7 | 10.6 | 2.6 | 7.0 | 25.1 | 20.2 | 16.8 | 22.8 | 28.7 | 29.8 | 13.9 | 26.1 | 22.6 | 17.5 | 15.4 | 19.1 |
| *Closed Source Models* | | | | | | | | | | | | | | | | | | | | | | | | | | | | |
| GPT-4o [40] | 75.4 | 61.6 | 32.3 | 62.5 | 97.3 | 96.1 | 47.1 | 87.4 | 100.0 | 96.3 | 30.0 | 81.9 | 79.6 | 71.7 | 29.4 | 68.9 | 56.8 | 37.0 | 17.6 | 47.3 | 90.8 | 68.1 | 42.9 | 71.6 | 95.0 | 66.7 | 41.2 | 72.3 |
| GPT-4o (+CoT) | 75.5 | 64.6 | 30.9 | 63.5 | 94.5 | 96.1 | 44.1 | 85.8 | 100.0 | 96.3 | 35.0 | 83.1 | 83.7 | 81.1 | 29.4 | 74.8 | 63.1 | 24.1 | 17.6 | 47.3 | 87.7 | 79.7 | 54.3 | 77.5 | 98.3 | 70.4 | 44.1 | 75.7 |
| Gemini-1.5-pro [49] | 71.8 | 57.4 | 26.9 | 58.3 | 93.2 | 90.8 | 32.4 | 80.9 | 100.0 | 96.3 | 20.0 | 79.5 | 71.4 | 79.2 | 23.5 | 68.1 | 60.4 | 18.5 | 23.5 | 44.5 | 93.8 | 58.0 | 22.9 | 64.5 | 91.7 | 81.5 | 32.4 | 74.3 |
| Gemini-1.5-pro (+CoT) | 70.8 | 58.6 | 27.0 | 58.4 | 90.4 | 92.1 | 29.4 | 79.8 | 100.0 | 85.2 | 25.0 | 77.1 | 81.6 | 79.2 | 11.8 | 70.6 | 61.3 | 27.8 | 23.5 | 47.8 | 92.3 | 59.4 | 40.0 | 68.0 | 95.0 | 79.6 | 26.5 | 73.6 |
| *Open Source Models* | | | | | | | | | | | | | | | | | | | | | | | | | | | | |
| LLaVA-NeXT (7B) [29] | 36.4 | 23.8 | 15.0 | 27.6 | 68.5 | 46.1 | 8.8 | 48.1 | 66.7 | 59.3 | 15.0 | 51.8 | 16.3 | 11.3 | 5.9 | 12.6 | 28.8 | 20.4 | 17.6 | 25.3 | 50.8 | 33.3 | 5.7 | 34.3 | 31.7 | 44.4 | 26.5 | 35.1 |
| LLaVA-NeXT (13B) | 41.1 | 28.6 | 16.4 | 31.8 | 79.5 | 53.9 | 8.8 | 55.7 | 75.0 | 74.1 | 10.0 | 59.0 | 28.6 | 32.1 | 11.8 | 27.7 | 22.5 | 27.8 | 11.8 | 23.1 | 73.8 | 43.5 | 11.4 | 48.5 | 35.0 | 40.7 | 14.7 | 32.4 |
| LLaVA-OneVision (7B) [26] | 51.0 | 37.8 | 18.1 | 40.0 | 79.5 | 69.7 | 23.5 | 65.0 | 94.4 | 85.2 | 15.0 | 72.3 | 36.7 | 20.8 | 5.9 | 25.8 | 28.8 | 22.2 | 17.6 | 25.8 | 78.5 | 40.6 | 20.0 | 50.9 | 65.0 | 42.6 | 8.8 | 43.9 |
| Table-LLaVA (7B) [60] | 32.5 | 24.3 | 13.5 | 25.9 | 53.4 | 26.3 | 5.9 | 33.3 | 47.2 | 63.0 | 20.0 | 45.8 | 20.4 | 11.3 | 5.9 | 14.3 | 22.5 | 25.9 | 11.8 | 22.5 | 41.5 | 26.1 | 11.4 | 29.0 | 25.0 | 61.1 | 26.5 | 38.5 |
| Math-LLaVA (13B) [47] | 37.7 | 27.1 | 15.7 | 29.6 | 54.8 | 38.2 | 8.8 | 39.3 | 50.0 | 70.4 | 15.0 | 48.2 | 16.3 | 24.5 | 17.6 | 20.2 | 17.1 | 18.5 | 11.8 | 15.8 | 67.7 | 33.3 | 17.1 | 43.2 | 33.3 | 38.9 | 11.8 | 30.4 |
| Phi-3.5-Vision-Instruct (4B) [1] | 49.0 | 34.8 | 16.5 | 37.7 | 78.1 | 51.3 | 8.8 | 54.1 | 86.1 | 70.4 | 10.0 | 62.7 | 38.8 | 47.2 | 11.8 | 38.7 | 22.5 | 33.3 | 23.5 | 25.8 | 86.2 | 39.1 | 25.7 | 54.4 | 51.7 | 38.9 | 11.8 | 37.8 |
| InternVL2 (8B) [10] | 43.0 | 31.6 | 13.3 | 33.5 | 65.8 | 52.6 | 8.8 | 49.1 | 30.6 | 63.0 | 15.0 | 37.3 | 32.7 | 30.2 | 17.6 | 29.4 | 28.8 | 18.5 | 0.0 | 23.1 | 60.0 | 30.4 | 11.4 | 37.9 | 45.0 | 40.7 | 0.0 | 33.1 |
| DeepSeek-VL (7B) [31] | 45.9 | 30.2 | 15.1 | 34.3 | 69.9 | 50.0 | 17.6 | 51.9 | 91.7 | 88.9 | 25.0 | 74.7 | 20.4 | 22.6 | 5.9 | 19.3 | 32.4 | 22.2 | 17.6 | 28.0 | 66.2 | 42.0 | 14.3 | 45.6 | 56.7 | 42.6 | 0.0 | 38.5 |

**Skills block 2: Cardinal, Cardinal Direction, Color, Congruence, Connectedness, Convexity, Coordinate**

| | Cardinal | | | | Cardinal Direction | | | | Color | | | | Congruence | | | | Connectedness | | | | Convexity | | | | Coordinate | | | |
|---|---|---|---|---|---|---|---|---|---|---|---|---|---|---|---|---|---|---|---|---|---|---|---|---|---|---|---|---|
| | easy | medium | hard | overall | easy | medium | hard | overall | easy | medium | hard | overall | easy | medium | hard | overall | easy | medium | hard | overall | easy | medium | hard | overall | easy | medium | hard | overall |
| Random Chance | 5.4 | 3.1 | 0.0 | 3.3 | 27.6 | 14.9 | 13.5 | 20.6 | 19.9 | 18.8 | 13.3 | 18.6 | 25.1 | 25.2 | 15.1 | 23.1 | 8.1 | 11.5 | 8.0 | 9.7 | 37.7 | 22.7 | 17.8 | 30.6 | 16.4 | 17.6 | 9.3 | 16.3 |
| *Closed Source Models* | | | | | | | | | | | | | | | | | | | | | | | | | | | | |
| GPT-4o | 84.4 | 66.7 | 22.7 | 64.0 | 86.8 | 93.8 | 25.0 | 80.5 | 90.4 | 85.7 | 46.9 | 82.7 | 81.5 | 26.9 | 31.8 | 49.8 | 71.2 | 67.1 | 25.0 | 60.7 | 82.3 | 50.0 | 69.2 | 70.6 | 71.7 | 52.9 | 21.4 | 56.9 |
| GPT-4o (+CoT) | 88.3 | 73.1 | 31.8 | 70.1 | 86.8 | 93.8 | 41.7 | 82.9 | 91.3 | 84.5 | 46.9 | 82.7 | 79.0 | 26.9 | 20.5 | 46.3 | 53.8 | 48.6 | 28.6 | 46.7 | 85.5 | 55.9 | 38.5 | 70.6 | 67.9 | 65.7 | 21.4 | 62.0 |
| Gemini-1.5-pro | 84.4 | 62.4 | 36.4 | 65.0 | 92.1 | 78.1 | 41.7 | 79.3 | 87.8 | 79.8 | 40.6 | 78.4 | 67.9 | 34.6 | 25.0 | 45.8 | 59.6 | 55.7 | 21.4 | 50.7 | 66.1 | 41.2 | 53.8 | 56.9 | 77.4 | 57.1 | 21.4 | 61.3 |
| Gemini-1.5-pro (+CoT) | 83.1 | 59.1 | 25.0 | 60.7 | 92.1 | 75.0 | 33.3 | 76.8 | 85.2 | 81.0 | 56.2 | 79.7 | 66.7 | 34.6 | 18.2 | 43.8 | 69.2 | 60.0 | 25.0 | 56.7 | 61.3 | 44.1 | 53.8 | 55.0 | 83.0 | 61.4 | 7.1 | 64.2 |
| *Open Source Models* | | | | | | | | | | | | | | | | | | | | | | | | | | | | |
| LLaVA-NeXT (7B) | 54.5 | 23.7 | 6.8 | 31.3 | 52.6 | 28.1 | 25.0 | 39.0 | 58.3 | 32.1 | 12.5 | 42.4 | 28.4 | 21.8 | 18.2 | 23.6 | 36.5 | 5.7 | 14.3 | 18.0 | 48.4 | 32.4 | 30.8 | 41.3 | 28.3 | 20.0 | 14.3 | 22.6 |
| LLaVA-NeXT (13B) | 66.2 | 25.8 | 9.1 | 36.9 | 57.9 | 37.5 | 50.0 | 48.8 | 64.3 | 41.7 | 21.9 | 50.2 | 29.6 | 16.7 | 11.4 | 20.7 | 50.0 | 21.4 | 21.4 | 31.3 | 50.0 | 35.3 | 23.1 | 42.2 | 22.6 | 24.3 | 14.3 | 22.6 |
| LLaVA-OneVision (7B) | 70.1 | 39.8 | 43.9 | 52.3 | 60.5 | 50.0 | 25.0 | 51.2 | 81.7 | 61.9 | 18.8 | 65.8 | 30.9 | 25.6 | 9.1 | 24.1 | 50.0 | 28.6 | 17.9 | 34.0 | 54.8 | 35.3 | 38.5 | 46.8 | 35.7 | 24.3 | 14.3 | 28.5 |
| Table-LLaVA (7B) | 41.6 | 15.1 | 6.8 | 22.9 | 44.7 | 25.0 | 8.3 | 31.7 | 49.6 | 28.6 | 18.8 | 37.7 | 17.3 | 24.1 | 11.4 | 20.7 | 32.7 | 11.4 | 25.0 | 21.3 | 41.9 | 35.3 | 15.4 | 36.7 | 15.1 | 21.4 | 0.0 | 16.8 |
| Math-LLaVA (13B) | 54.5 | 29.0 | 6.8 | 33.6 | 52.6 | 31.2 | 25.0 | 40.2 | 64.3 | 27.4 | 15.6 | 44.2 | 25.9 | 19.2 | 11.4 | 20.2 | 34.6 | 8.6 | 17.9 | 19.3 | 51.6 | 23.5 | 15.4 | 38.5 | 22.6 | 25.7 | 21.4 | 24.1 |
| Phi-3.5-Vision-Instruct (4B) | 62.3 | 34.4 | 9.1 | 39.3 | 73.7 | 53.1 | 0.0 | 54.9 | 75.7 | 57.1 | 15.6 | 60.6 | 29.6 | 24.4 | 18.2 | 25.1 | 26.0 | 34.3 | 21.4 | 28.7 | 43.5 | 44.1 | 46.2 | 44.0 | 28.3 | 34.3 | 21.4 | 30.7 |
| InternVL2 (8B) | 50.6 | 18.3 | 6.8 | 27.6 | 50.0 | 40.6 | 25.0 | 42.7 | 65.2 | 39.3 | 12.5 | 48.5 | 29.6 | 16.7 | 27.3 | 24.1 | 36.5 | 22.9 | 21.4 | 27.3 | 54.8 | 23.5 | 7.7 | 39.4 | 24.5 | 35.7 | 14.3 | 29.2 |
| DeepSeek-VL (7B) | 66.2 | 31.2 | 13.6 | 40.2 | 63.2 | 37.5 | 33.3 | 48.8 | 68.7 | 38.1 | 12.5 | 49.8 | 30.9 | 15.4 | 13.6 | 21.2 | 28.8 | 31.4 | 28.8 | 29.3 | 51.6 | 32.4 | 30.8 | 43.1 | 18.9 | 27.1 | 0.0 | 21.2 |

**Skills block 3: Curvature, Direction, Interior, Intersection, Length, Line, Overlap**

| | Curvature | | | | Direction | | | | Interior | | | | Intersection | | | | Length | | | | Line | | | | Overlap | | | |
|---|---|---|---|---|---|---|---|---|---|---|---|---|---|---|---|---|---|---|---|---|---|---|---|---|---|---|---|---|
| | easy | medium | hard | overall | easy | medium | hard | overall | easy | medium | hard | overall | easy | medium | hard | overall | easy | medium | hard | overall | easy | medium | hard | overall | easy | medium | hard | overall |
| Random Chance | 26.2 | 27.3 | 16.7 | 24.2 | 23.3 | 17.2 | 11.8 | 17.9 | 23.7 | 15.0 | 17.9 | 19.2 | 28.8 | 24.9 | 8.2 | 24.6 | 31.4 | 21.5 | 17.1 | 24.5 | 20.5 | 12.9 | 5.3 | 16.6 | 24.4 | 26.9 | 2.3 | 21.6 |
| *Closed Source Models* | | | | | | | | | | | | | | | | | | | | | | | | | | | | |
| GPT-4o | 84.6 | 65.5 | 67.6 | 71.8 | 86.8 | 51.3 | 16.7 | 55.1 | 93.9 | 79.7 | 33.3 | 80.4 | 64.3 | 50.0 | 25.0 | 52.4 | 72.9 | 69.4 | 40.0 | 64.8 | 76.4 | 60.5 | 90.0 | 72.0 | 92.9 | 71.2 | 29.2 | 72.7 |
| GPT-4o (+CoT) | 89.7 | 65.5 | 64.7 | 72.5 | 86.8 | 60.5 | 12.5 | 59.4 | 92.4 | 84.4 | 44.4 | 83.1 | 61.9 | 51.4 | 33.3 | 53.2 | 70.8 | 65.3 | 36.0 | 61.5 | 75.0 | 79.1 | 80.0 | 76.8 | 89.3 | 76.3 | 25.0 | 72.7 |
| Gemini-1.5-pro | 82.1 | 41.4 | 38.2 | 52.7 | 81.6 | 51.3 | 16.7 | 53.6 | 92.4 | 81.2 | 33.3 | 80.4 | 61.9 | 50.0 | 25.0 | 51.6 | 66.7 | 65.3 | 44.0 | 61.5 | 80.6 | 55.8 | 70.0 | 71.2 | 83.9 | 71.2 | 12.5 | 66.2 |
| Gemini-1.5-pro (+CoT) | 84.6 | 39.7 | 38.2 | 52.7 | 89.5 | 50.0 | 8.3 | 53.6 | 95.5 | 79.7 | 33.3 | 81.1 | 64.3 | 54.2 | 25.0 | 54.8 | 64.6 | 57.1 | 32.0 | 54.9 | 79.2 | 79.1 | 80.0 | 79.2 | 83.9 | 69.5 | 25.0 | 67.6 |
| *Open Source Models* | | | | | | | | | | | | | | | | | | | | | | | | | | | | |
| LLaVA-NeXT (7B) | 12.8 | 29.3 | 20.6 | 22.1 | 39.5 | 26.3 | 12.5 | 27.5 | 53.0 | 35.9 | 11.1 | 40.5 | 38.1 | 26.4 | 16.7 | 29.4 | 45.8 | 36.7 | 24.0 | 37.7 | 26.4 | 18.6 | 20.0 | 23.2 | 10.7 | 20.3 | 12.5 | 15.1 |
| LLaVA-NeXT (13B) | 23.1 | 29.3 | 17.6 | 24.4 | 39.5 | 14.5 | 29.2 | 23.9 | 48.5 | 37.5 | 22.2 | 40.5 | 19.0 | 19.4 | 25.0 | 19.8 | 45.8 | 40.8 | 20.0 | 38.5 | 37.2 | 10.0 | 35.3 | 33.9 | 20.3 | 35.7 | 8.3 | 30.2 |
| LLaVA-OneVision (7B) | 66.7 | 34.5 | 17.6 | 39.7 | 68.4 | 31.6 | 29.2 | 41.3 | 71.2 | 56.2 | 38.9 | 60.8 | 33.3 | 20.8 | 0.0 | 23.0 | 52.1 | 49.0 | 16.0 | 43.4 | 61.1 | 37.2 | 50.0 | 52.0 | 41.1 | 33.9 | 12.5 | 33.1 |
| Table-LLaVA (7B) | 17.9 | 22.4 | 14.7 | 19.1 | 28.9 | 15.8 | 12.5 | 18.8 | 48.5 | 32.8 | 33.3 | 39.9 | 26.2 | 23.6 | 25.0 | 24.6 | 45.8 | 32.7 | 12.0 | 33.6 | 18.1 | 11.6 | 10.0 | 15.2 | 12.5 | 13.6 | 4.2 | 11.5 |
| Math-LLaVA (13B) | 20.5 | 25.9 | 20.6 | 22.1 | 36.8 | 25.0 | 12.5 | 26.1 | 43.9 | 31.2 | 16.7 | 35.1 | 31.0 | 20.8 | 16.7 | 23.8 | 45.8 | 42.9 | 0.0 | 39.3 | 18.6 | 10.0 | 24.0 | 18.6 | 44.6 | 23.7 | 16.7 | 30.9 |
| Phi-3.5-Vision-Instruct (4B) | 61.5 | 29.3 | 17.6 | 35.9 | 55.3 | 32.9 | 12.5 | 35.5 | 59.1 | 35.9 | 38.9 | 46.6 | 45.2 | 23.6 | 16.7 | 30.2 | 54.2 | 55.1 | 16.0 | 46.7 | 32.6 | 30.0 | 44.8 | 41.1 | 28.8 | 25.0 | 8.3 | 30.2 |
| InternVL2 (8B) | 43.6 | 24.1 | 8.8 | 26.0 | 50.0 | 26.3 | 8.3 | 29.7 | 68.2 | 43.8 | 11.1 | 50.7 | 33.3 | 25.0 | 16.7 | 27.0 | 45.8 | 51.0 | 8.0 | 40.2 | 40.3 | 30.2 | 30.0 | 36.0 | 46.4 | 39.0 | 16.7 | 38.1 |
| DeepSeek-VL (7B) | 56.4 | 36.2 | 17.6 | 37.4 | 60.5 | 22.4 | 16.7 | 31.9 | 59.1 | 46.9 | 16.7 | 48.6 | 38.1 | 31.9 | 16.7 | 32.5 | 39.6 | 40.8 | 8.0 | 40.8 | 50.0 | 20.9 | 20.0 | 37.6 | 48.2 | 16.9 | 12.5 | 28.8 |

**Skills block 4: Ordinal, Orientation, Orthogonality, Parallel, Point, Reflection, Relative Position, Rotation**

| | Ordinal | | | | Orientation | | | | Orthogonality | | | | Parallel | | | | Point | | | | Reflection | | | | Relative Position | | | | Rotation | | | |
|---|---|---|---|---|---|---|---|---|---|---|---|---|---|---|---|---|---|---|---|---|---|---|---|---|---|---|---|---|---|---|---|---|
| | easy | medium | hard | overall | easy | medium | hard | overall | easy | medium | hard | overall | easy | medium | hard | overall | easy | medium | hard | overall | easy | medium | hard | overall | easy | medium | hard | overall | easy | medium | hard | overall |
| Random Chance | 23.3 | 16.5 | 2.7 | 15.4 | 27.6 | 33.0 | 26.0 | 29.2 | 24.4 | 13.5 | 13.7 | 18.3 | 24.3 | 22.0 | 18.7 | 22.4 | 24.7 | 20.3 | 3.9 | 18.8 | 26.7 | 25.2 | 17.6 | 24.8 | 28.3 | 26.4 | 18.0 | 26.3 | 32.9 | 27.9 | 15.5 | 24.8 |
| *Closed Source Models* | | | | | | | | | | | | | | | | | | | | | | | | | | | | | | | | |
| GPT-4o | 89.1 | 78.2 | 30.3 | 70.1 | 42.4 | 42.9 | 30.0 | 40.5 | 60.0 | 48.4 | 21.4 | 48.8 | 41.1 | 34.9 | 16.1 | 34.0 | 92.5 | 87.1 | 25.0 | 75.9 | 53.2 | 37.1 | 20.0 | 41.2 | 83.3 | 50.0 | 40.0 | 65.5 | 37.1 | 21.1 | 18.0 | 23.9 |
| GPT-4o (+CoT) | 97.8 | 85.5 | 30.3 | 76.1 | 50.8 | 50.0 | 40.0 | 48.8 | 62.9 | 51.6 | 21.4 | 51.2 | 50.7 | 51.2 | 6.5 | 41.5 | 96.2 | 96.8 | 33.3 | 82.4 | 39.0 | 39.2 | 20.0 | 36.7 | 88.1 | 46.9 | 30.0 | 65.5 | 37.2 | 31.6 | 10.0 | 21.8 |
| Gemini-1.5-pro | 78.3 | 60.0 | 12.1 | 54.5 | 50.8 | 23.8 | 30.0 | 38.0 | 60.0 | 38.7 | 35.7 | 47.5 | 34.2 | 53.5 | 3.2 | 33.3 | 96.2 | 96.8 | 50.0 | 86.1 | 46.8 | 20.6 | 20.0 | 30.7 | 61.9 | 31.2 | 30.0 | 46.4 | 40.0 | 17.5 | 16.0 | 22.5 |
| Gemini-1.5-pro (+CoT) | 89.1 | 85.5 | 21.2 | 70.9 | 39.0 | 38.1 | 40.0 | 38.8 | 62.9 | 29.0 | 21.4 | 42.5 | 27.4 | 67.4 | 19.4 | 37.4 | 96.2 | 90.3 | 54.2 | 85.2 | 35.1 | 23.7 | 16.0 | 27.1 | 57.1 | 34.4 | 30.0 | 45.2 | 31.4 | 10.5 | 10.0 | 15.5 |
| *Open Source Models* | | | | | | | | | | | | | | | | | | | | | | | | | | | | | | | | |
| LLaVA-NeXT (7B) | 28.3 | 12.7 | 9.1 | 17.2 | 27.1 | 23.8 | 35.0 | 27.3 | 28.6 | 12.9 | 7.1 | 18.8 | 11.0 | 18.6 | 12.9 | 13.6 | 45.3 | 22.6 | 0.0 | 28.7 | 22.1 | 7.2 | 32.0 | 16.1 | 31.0 | 28.1 | 0.0 | 26.2 | 22.9 | 17.5 | 16.0 | 18.3 |
| LLaVA-NeXT (13B) | 43.5 | 18.2 | 15.2 | 26.1 | 28.8 | 35.7 | 50.0 | 34.7 | 17.1 | 16.1 | 28.6 | 18.8 | 12.3 | 9.3 | 19.4 | 12.9 | 41.5 | 35.5 | 4.2 | 31.5 | 28.6 | 20.6 | 16.0 | 23.1 | 47.6 | 31.2 | 0.0 | 35.7 | 46.3 | 14.0 | 20.0 | 14.8 |
| LLaVA-OneVision (7B) | 52.2 | 21.8 | 12.1 | 29.9 | 33.9 | 26.2 | 35.0 | 31.4 | 37.1 | 12.9 | 7.1 | 22.5 | 5.5 | 18.6 | 16.1 | 11.6 | 47.2 | 54.8 | 4.2 | 39.8 | 35.1 | 38.1 | 16.0 | 34.2 | 50.0 | 40.6 | 20.0 | 42.9 | 40.0 | 22.8 | 24.0 | 27.5 |
| Table-LLaVA (7B) | 34.8 | 25.5 | 3.0 | 23.2 | 26.2 | 26.2 | 30.0 | 29.8 | 31.4 | 29.0 | 0.0 | 25.0 | 12.3 | 20.9 | 16.1 | 15.6 | 34.0 | 32.3 | 8.3 | 27.8 | 27.3 | 14.4 | 4.0 | 18.1 | 38.1 | 18.8 | 10.0 | 27.4 | 40.0 | 24.6 | 22.0 | 27.5 |
| Math-LLaVA (13B) | 30.4 | 16.4 | 3.0 | 17.9 | 37.3 | 31.0 | 50.0 | 37.2 | 25.7 | 29.0 | 0.0 | 22.5 | 21.9 | 20.9 | 25.8 | 22.4 | 45.3 | 38.7 | 0.0 | 33.3 | 32.5 | 18.6 | 16.0 | 23.6 | 38.1 | 28.1 | 20.0 | 32.1 | 51.4 | 33.3 | 18.0 | 32.4 |
| Phi-3.5-Vision-Instruct (4B) | 54.3 | 38.2 | 21.2 | 39.6 | 32.2 | 42.9 | 40.0 | 37.2 | 28.6 | 12.9 | 28.6 | 22.5 | 27.4 | 30.2 | 16.1 | 25.9 | 43.4 | 45.2 | 0.0 | 34.3 | 44.2 | 29.9 | 4.0 | 32.2 | 42.9 | 30.6 | 30.0 | 37.7 | 54.3 | 3.5 | 20.0 | 21.8 |
| InternVL2 (8B) | 41.3 | 40.0 | 6.1 | 32.1 | 28.8 | 35.7 | 40.0 | 33.1 | 48.6 | 6.5 | 14.3 | 26.2 | 17.8 | 30.2 | 16.1 | 21.1 | 49.1 | 51.6 | 0.0 | 38.9 | 35.1 | 29.9 | 20.0 | 30.7 | 40.5 | 25.0 | 10.0 | 31.0 | 25.7 | 26.3 | 10.0 | 20.4 |
| DeepSeek-VL (7B) | 28.3 | 21.6 | 6.1 | 21.6 | 32.2 | 19.0 | 45.0 | 29.8 | 25.7 | 22.6 | 21.4 | 23.8 | 2.7 | 16.3 | 19.4 | 10.2 | 52.8 | 71.0 | 4.2 | 47.2 | 36.4 | 13.4 | 16.0 | 22.6 | 40.5 | 43.8 | 10.0 | 38.1 | 11.4 | 15.8 | 12.0 | 13.4 |

**Skills block 5: Rotational Symmetry, Shape, Sharpness, Similarity, Symbol, Tangency, Texture, Width**

| | Rotational Symmetry | | | | Shape | | | | Sharpness | | | | Similarity | | | | Symbol | | | | Tangency | | | | Texture | | | | Width | | | |
|---|---|---|---|---|---|---|---|---|---|---|---|---|---|---|---|---|---|---|---|---|---|---|---|---|---|---|---|---|---|---|---|---|
| | easy | medium | hard | total | easy | medium | hard | total | easy | medium | hard | total | easy | medium | hard | total | easy | medium | hard | total | easy | medium | hard | total | easy | medium | hard | total | easy | medium | hard | total |
| Random Chance | 28.0 | 24.3 | 15.1 | 24.5 | 11.7 | 6.7 | 17.9 | 9.7 | 21.7 | 17.8 | 12.7 | 18.6 | 24.6 | 18.7 | 13.3 | 19.8 | 21.0 | 18.6 | 3.8 | 17.1 | 27.7 | 24.2 | 20.8 | 25.9 | 23.2 | 15.8 | 2.3 | 18.1 | 25.0 | 22.3 | 13.5 | 21.9 |
| *Closed Source Models* | | | | | | | | | | | | | | | | | | | | | | | | | | | | | | | | |
| GPT-4o | 61.0 | 69.0 | 21.4 | 61.1 | 98.1 | 83.8 | 50.0 | 85.7 | 85.5 | 83.8 | 25.9 | 72.2 | 86.0 | 47.4 | 27.3 | 57.4 | 76.0 | 78.6 | 14.3 | 66.9 | 24.7 | 24.6 | 26.7 | 24.8 | 83.7 | 71.4 | 33.3 | 73.1 | 72.2 | 60.0 | 39.1 | 61.1 |
| GPT-4o (+CoT) | 68.3 | 77.5 | 7.1 | 66.7 | 98.1 | 83.8 | 64.3 | 87.1 | 88.7 | 75.7 | 18.5 | 69.8 | 69.8 | 56.1 | 9.1 | 52.5 | 74.0 | 78.6 | 0.0 | 63.8 | 30.1 | 21.1 | 13.3 | 25.5 | 81.4 | 71.4 | 42.9 | 73.1 | 75.9 | 60.0 | 39.1 | 62.4 |
| Gemini-1.5-pro | 46.3 | 64.8 | 7.1 | 52.4 | 98.1 | 74.3 | 50.0 | 80.7 | 83.9 | 83.8 | 18.5 | 69.8 | 67.4 | 45.6 | 13.6 | 45.1 | 76.0 | 80.4 | 0.0 | 65.4 | 36.6 | 36.8 | 40.0 | 37.0 | 70.9 | 69.4 | 47.6 | 67.3 | 55.6 | 48.8 | 21.7 | 49.0 |
| Gemini-1.5-pro (+CoT) | 43.9 | 62.0 | 14.3 | 50.8 | 96.2 | 78.4 | 57.1 | 82.9 | 77.4 | 73.0 | 25.9 | 65.1 | 62.8 | 43.9 | 13.6 | 45.1 | 80.0 | 83.9 | 0.0 | 68.5 | 31.2 | 28.1 | 13.3 | 28.5 | 70.9 | 69.4 | 42.9 | 66.7 | 55.6 | 44.8 | 21.7 | 47.1 |
| *Open Source Models* | | | | | | | | | | | | | | | | | | | | | | | | | | | | | | | | |
| LLaVA-NeXT (7B) | 34.1 | 12.7 | 7.1 | 19.0 | 57.7 | 23.0 | 28.6 | 36.4 | 33.9 | 16.2 | 7.4 | 23.0 | 20.9 | 10.5 | 13.6 | 14.8 | 40.0 | 32.1 | 4.8 | 30.7 | 29.0 | 19.3 | 20.0 | 24.8 | 34.9 | 18.4 | 4.8 | 25.6 | 31.5 | 27.5 | 34.8 | 29.9 |
| LLaVA-NeXT (13B) | 31.7 | 2.8 | 14.3 | 13.5 | 67.3 | 45.9 | 21.4 | 51.4 | 46.8 | 24.3 | 7.4 | 31.7 | 37.2 | 14.0 | 4.5 | 20.5 | 38.0 | 35.7 | 9.5 | 32.3 | 24.7 | 14.0 | 26.7 | 21.2 | 38.4 | 32.7 | 0.0 | 31.4 | 46.3 | 26.2 | 30.4 | 33.8 |
| LLaVA-OneVision (7B) | 56.1 | 32.4 | 7.1 | 37.3 | 92.3 | 66.2 | 42.9 | 73.6 | 53.2 | 37.8 | 25.9 | 42.9 | 23.3 | 8.8 | 36.4 | 18.9 | 64.0 | 76.8 | 14.3 | 61.4 | 28.0 | 10.5 | 20.0 | 21.2 | 36.0 | 40.8 | 4.8 | 33.3 | 33.3 | 35.0 | 21.7 | 32.5 |
| Table-LLaVA (7B) | 48.8 | 14.1 | 7.1 | 24.6 | 38.5 | 25.7 | 21.4 | 30.0 | 32.3 | 21.6 | 14.8 | 25.4 | 16.3 | 14.0 | 9.1 | 13.9 | 34.0 | 42.9 | 4.8 | 34.6 | 33.3 | 15.8 | 20.0 | 26.7 | 31.4 | 18.4 | 0.0 | 23.1 | 33.3 | 35.0 | 21.7 | 32.5 |
| Math-LLaVA (13B) | 31.7 | 8.5 | 35.7 | 19.0 | 69.2 | 37.8 | 35.7 | 49.3 | 41.9 | 21.6 | 29.6 | 33.3 | 9.3 | 12.3 | 4.5 | 9.8 | 26.0 | 51.8 | 4.8 | 33.9 | 23.7 | 24.6 | 26.7 | 24.2 | 31.4 | 10.2 | 9.5 | 21.8 | 35.2 | 43.8 | 8.7 | 35.7 |
| Phi-3.5-Vision-Instruct (4B) | 46.3 | 12.7 | 0.0 | 22.2 | 82.7 | 45.9 | 35.7 | 58.6 | 46.8 | 27.0 | 3.7 | 31.7 | 37.2 | 21.1 | 18.2 | 26.2 | 42.0 | 55.4 | 9.5 | 42.5 | 29.0 | 28.1 | 20.0 | 27.9 | 55.8 | 12.2 | 9.5 | 35.3 | 40.7 | 31.2 | 17.4 | 32.5 |
| InternVL2 (8B) | 65.9 | 12.7 | 28.6 | 31.7 | 80.8 | 50.0 | 28.6 | 59.3 | 50.0 | 24.3 | 18.5 | 35.7 | 16.3 | 12.3 | 18.2 | 14.8 | 36.0 | 48.2 | 0.0 | 35.4 | 34.4 | 21.1 | 20.0 | 28.5 | 37.2 | 38.8 | 4.8 | 33.3 | 29.6 | 32.5 | 17.4 | 29.3 |
| DeepSeek-VL (7B) | 41.5 | 4.2 | 28.6 | 19.0 | 80.8 | 45.9 | 42.9 | 58.6 | 29.7 | 3.7 | 46.8 | 32.6 | 18.2 | 21.1 | 18.2 | 24.6 | 52.0 | 53.6 | 9.5 | 45.7 | 30.1 | 12.3 | 26.7 | 23.6 | 41.9 | 20.4 | 0.0 | 29.5 | 44.4 | 30.0 | 13.0 | 32.5 |

