# OpenReview forum: "Decomposing Complex Visual Comprehension into Atomic Visual Skills for Vision Language Models"
_NeurIPS.cc/2024/Workshop/MATH-AI — MATH-AI 24_

### Official Review · Reviewer_BBdT · 2024-10-01
**Review of paper: Decomposing Complex Visual Comprehension into Atomic Visual Skills for Vision Language Models**

**Rating:** 5
**Confidence:** 3

**Review:**

# Review of paper: Decomposing Complex Visual Comprehension into Atomic Visual Skills for Vision Language Models

# Reviewer Summary of paper:

The paper proposes a new benchmark dataset named, AVSBench which consists of

# Major points:

1. Hypothesis 1 is not directly answered in the work as far as I can tell and only indrectly answered via the results. While I appreciate the simplicity of using problems with a single skill, the authors should provide some descriptions of why multi-skill tasks were not included.

2. MathVerse found that models use shortcuts in language only portion of models but also found
mathematical training benefits performance, cf. Section 3.2 of the paper.
Moreover, the majority of the models did not design the neural network architecture for domain specific multimodal language models like math problems.
In "Smart Vision-Language Reasoners", also presented at ICML 2024 in tandem with Mathverse, the authors found that designing and fine tuning for domain specific multimodal language models improved performance on the Smart-101 dataset, an image + text dataset which is similar in structure to the AVSBench data.
Specifically, on the architecture aspect of the multimodal language models in "Smart Vision-Language Reasoners" they introduced via a cross attention layer to between the textual backbone and the vision backbone model to promote alignment of visual and textual features.
This is in Contrast with the Llava architecture which uses a learned projection layer and an MLP to mix the weights of the vision backbone (CLIP) and the text backbone (Vicuna), cf. section 4.1 of the original Llava paper.

Suggest to reference these two papers:

Are Deep Neural Networks SMARTer than Second Graders? CVPR 2023

https://openaccess.thecvf.com/content/CVPR2023/papers/Cherian_Are_Deep_Neural_Networks_SMARTer_Than_Second_Graders_CVPR_2023_paper.pdf

Smart Vision-Language Reasoners, ICML Math AI workshop 2024

https://arxiv.org/abs/2407.04212

Note: The Smart-101 dataset focuses on problems for children ages 6-8 so differ from those of high school geometry.

3. The authors mention that the skill difficulties for each problem are subjectively determined by the authors themselves. However, no mechanism, e.g. other annotators were used. Recommend the authors use independent annotators or use the multimodal languages evaluated in the manuscript to confirm (or not) that these difficulties match.

4. The skill categories are claimed to be fundamental/atomic for geometric problem solving but there are no references to indicate the source of the taxonomy nor support claims of being fundamental skills for geometry problems.

# Minor points:

1. The author(s) claim without reference that the curricula of high school geometry are more or less agreed upon. The claim is made without a reference, after searching the PISA site and using a major search engine, I was unable to find something that showed a generally agreed upon high school geometry syllabus.

The referee suggests to rephrase as "minimal common curricular elements" or something similar, or to provide a reference to the commonly agreed sylabus.

2. The referee did a quick search engine check and AVSBench is a previously used name for an audio visual benchmark dataset, cf. https://paperswithcode.com/dataset/avsbench, perhaps a different name should be chosen to avoid namespace collisions?

# Additional remarks:

1. In addition to the, COT prompting, the authors might try visual-COT which has been proposed in the literature to compare with pure textual COT.

---

### Official Review · Reviewer_REby · 2024-10-07
**Review of AVS for VLMs**

**Rating:** 8
**Confidence:** 5

**Review:**

This work focuses on evaluating VLMs' perception as understanding details in abstract shapes. The authors introduce a novel benchmark, that decomposes 36 different atomic skills for perception. The work evaluates existing VLMs and shows that these models struggle with such tasks. The work is interesting and significant for the field as it shows VLMs' shortcomings in perceiving abstract shapes in a more fine-grain way as opposed to previous work which had focused on the general performance of these models on such tasks.

I find these areas for improvement:
1) I am wondering if these skills are truly atomic. For instance, the "intersection" in Fig 1. is not an atomic task. I think there are at least two skills involved here: 1) the ability to find intersections, and 2) the ability to count found intersections. As we know about these models, they are not good at counting, so I assume a model might be able to find the interactions, but unable to count them (however, in the example provided in Fig 1., it is not the case because of the great distance between the correct answer and the one provided by GPT-4o). Overall, this issue concerns me about those questions that involve counting. I think authors can simply improve the reliability of their results by removing the questions that involve counting and more generally, looking for skills that are not truly atomic and removing them from AVSBench.

2) The models evaluated are very interesting as they show that specialized models such as Math-Llava also struggle with these tasks and CoT does not help that much. These results resonate with some other work that the authors might want to refer to as well ([1], [2], [3]). I am wondering if fine-tuning Llava might help with its performance. To further increase the significance of this work, it would be important if the authors evaluated a fine-tuned model that is specifically fine-tuned on these AVS skills (as mentioned in the conclusion section).

3) I find that maybe the most of atomic tasks are not high school level. For instance, all of the examples provided in Fig 1. are elementary level. It would be important if authors could provide fine-grained detail on the age appropriateness of these skills. Also, are the authors claiming that these skills are exclusive? If so, how did they make sure that they included all the atomic skills?

4) I find the evaluation setup a bit overly complicated. Why did the authors use another LLM (GPT mini) to extract the response generated by the VLM? Perhaps they could have had the models generate their response in a specific format so they could have parsed the output with no further API calls.

Overall, I find this work a great contribution to the MATH-AI workshop as it shows important shortcomings in VLMs that need to be overcome in the future. I like the writing quality of the paper and I believe that it can benefit from the above points to be in great shape for future conference submissions.


[1] Fu, D., Khalighinejad, G., Liu, O., Dhingra, B., Yogatama, D., Jia, R., & Neiswanger, W. (2024). IsoBench: Benchmarking Multimodal Foundation Models on Isomorphic Representations. arXiv preprint arXiv:2404.01266.
[2] Rismanchian, S., Doroudi, S., & Razeghi, Y. (2024, May). Turtle-like Geometry Learning: How Humans and Machines Differ in Learning Turtle Geometry. In Proceedings of the AAAI Symposium Series (Vol. 3, No. 1, pp. 586-587).
[3] Melanie Mitchell, Alessandro B Palmarini, and Arseny Moskvichev. Comparing humans, gpt-4, and gpt-4v on abstraction and reasoning tasks. arXiv preprint arXiv:2311.09247, 2023.

---

### Official Review · Reviewer_gKF2 · 2024-10-08

**Rating:** 7
**Confidence:** 4

**Review:**

**Summary:**
The paper presents the *Atomic Visual Skills Benchmark (AVSBench)*, a dataset designed to evaluate VLMs on their ability to comprehend basic geometric features, termed *atomic visual skills*. The benchmark includes 5,073 handcrafted image-question-answer pairs covering 36 skills, such as understanding angles, boundaries, and congruence. It aims to highlight the limitations of current VLMs, which struggle with these seemingly trivial tasks, despite their success in more complex scenarios. Models like GPT-4o and Gemini-1.5-pro perform relatively well on easier tasks but struggle with harder ones.

**Strengths:**
1. **Useful Dataset with Detailed Categories:** AVSBench provides a comprehensive and systematic breakdown of 36 atomic visual skills, covering a wide range of essential visual features relevant to high-school level geometry and other visual comprehension tasks. This detailed categorization enables more granular evaluation of VLM capabilities.

2. **Well-Motivated Problem:** The paper addresses an important issue—VLMs often fail at tasks involving simple visual comprehension. The problem is clearly defined, and the benchmark helps shed light on the underlying limitations of these models, thereby contributing to the understanding of VLM behavior.

3. **Comprehensive Models for Evaluation:** The paper evaluates several state-of-the-art VLMs, including GPT-4o and Gemini-1.5-pro, along with other open models. The results are compared across different levels of task difficulty, providing insights into how various models perform across the spectrum of atomic visual skills.

**Weaknesses:**
1. **Not Enough Baselines:** The study only explores the performance of the models with chain-of-thought (CoT) prompting and does not investigate other advanced text or visual prompting methods. Incorporating more diverse baselines would strengthen the evaluation and provide a broader perspective on how different techniques could improve model performance.

---

### Official Review · Reviewer_nA6r · 2024-10-08
**The paper is overall well-written and the main ideas are clearly explained.**

**Rating:** 5
**Confidence:** 2

**Review:**

Paper Summary:
This paper introduces AVSBench, a novel benchmark designed to evaluate 36 atomic visual skills in geometry and conduct experiments on influential models to show that they struggle with fundamental geometric visual tasks.

Weakness and Strength:

(+) The paper is well-structured and presents the problem and findings.

(+) This paper studies an interesting and well-motivated research topic: Benchmark to evaluate whether VLMs possess capabilities to understand basic geometric features.

(-) The introduction to the novel benchmark is somewhat limited.

---

### Decision · Program_Chairs · 2024-10-09

Accept